# Ring deconvolution microscopy: exploiting symmetry for efficient spatially varying aberration correction

Amit Kohli [1,7] ✉, Anastasios N. Angelopoulos [1,7], David McAllister[1], Esther Whang[2], Sixian You [3], Kyrollos Yanny[4], Federico M. Gasparoli [5], Bo-Jui Chang [6], Reto Fiolka [6] & Laura Waller [1] ✉

The most ubiquitous form of aberration correction for microscopy is deconvolution; however, deconvolution relies on the assumption that the system's point spread function is the same across the entire field of view. This assumption is often inadequate, but space-variant deblurring techniques generally require impractical amounts of calibration and computation. We present an imaging pipeline that leverages symmetry to provide simple and fast spatially varying deblurring. Our ring deconvolution microscopy method utilizes the rotational symmetry of most microscopes and cameras, and naturally extends to sheet deconvolution in the case of lateral symmetry. We derive theory and algorithms for ring deconvolution microscopy and propose a neural network based on Seidel aberration coefficients as a fast alternative. We demonstrate improvements in speed and image quality as compared to standard deconvolution and existing spatially varying deblurring across a diverse range of microscope modalities, including miniature microscopy, multicolor fluorescence microscopy, multimode fiber micro-endoscopy and light-sheet fluorescence microscopy. Our approach enables near-isotropic, subcellular resolution in each of these applications.

Much of optical engineering is focused on reducing aberrations by adding additional corrective optical elements to an imaging system; consider a microscope objective, consisting of numerous lenses stacked in a housing. Such designs allow for high-performance imaging, but incur added cost, weight and complication. Even with large and expensive lens stacks, it is difficult and, in some cases, impossible to correct all aberrations across a large area and so aberrations are often what limit the usable field of view (FoV) of a system. Furthermore, some systems cannot accommodate any aberration-correction optics; for example, additional elements may not fit in miniaturized microscopes[1] and are prohibitively expensive for large-aperture telescopes[2].

Faced with a poorly corrected imaging system, the modern microscopist instead turns to computational aberration correction, where the burden is shifted onto computer algorithms applied post-capture. The most commonly used correction technique, image deconvolution, captures a calibration image of a small point-like source, known as the point spread function (PSF), to characterize the aberrations. The PSF can then be used to computationally deconvolve any image taken with

[1]Department of Electrical Engineering and Computer Sciences, UC Berkeley, Berkeley, CA, USA. [2]Department of Electrical Engineering, The Cooper Union, New York, NY, USA. [3]Department of Electrical Engineering and Computer Science, MIT, Cambridge, MA, USA. [4]UCB/UCSF Joint Graduate Program in Bioengineering, UC Berkeley, Berkeley, CA, USA. [5]Department of Cell Biology, Harvard Medical School, Boston, MA, USA. [6]Lyda Hill Department of Bioinformatics, UT Southwestern Medical Center, Dallas, TX, USA. [7]These authors contributed equally: Amit Kohli, Anastasios N. Angelopoulos. ✉e-mail: apkohli@berkeley.edu; waller@berkeley.edu

the system via simple and fast algorithms, to yield a deblurred result. A main limitation of this approach is that it assumes that the system's PSF does not vary spatially (the system is linear space-invariant; LSI). This assumption is usually only true near the center of the FoV, and optical designers often artificially sacrifice part of the system's FoV to maintain space-invariance.

To go beyond space-invariant limitations, a large community effort has gone toward heuristic forms of spatially varying 'deconvolution', wherein one measures PSFs at multiple points within the FoV and uses them to correct the image. Such heuristics include assuming that each region of an image is locally LSI[3], adaptively splitting the FoV by first quantifying the degree of space-variance[4,5], interpolating PSFs[6,7], decomposing the PSF into space-invariant orthogonal modes[8–15], and doing the same in Fourier space[16]. These heuristics can approach rigorous recovery as the number of PSFs collected grows, possibly into the hundreds of thousands; however, the trade-off in terms of the complexity of calibration and computation quickly becomes intractable. For example, in patch-wise deconvolution, the FoV is divided into patches, each of which is deconvolved by a PSF measured at its center. Maximum accuracy is achieved when the patch size is reduced to a single pixel, but then a megapixel image would require a million PSF measurements and a computation time of hundreds of hours to deblur.

Another emerging modality is deep deblurring[17–23], in which varying amounts of system information are incorporated into a deep neural network. Networks that are primarily data-driven struggle with extrapolation beyond the training data, and tend to reproduce whatever biases existed therein, a particularly relevant point as many of them are trained on simulated data. Meanwhile, networks that incorporate physical information, such as calibrated PSFs, may have better generalization properties but suffer the same accuracy/efficiency trade-off as patch-based methods. For these reasons, spatially varying deblurring has not become commonplace among practitioners, and there remains a need for spatially varying deblurring methods that are effective, efficient and robust.

Here we propose a spatially varying method that requires only a single calibration image and has reasonable compute time, while offering rigorous deblurring for imaging systems that are symmetric in some way. We focus on rotationally symmetric systems (systems that are symmetric about their optical axis) but also show an example with the lateral symmetry present in light-sheet microscopy. Rotational symmetry occurs in many imaging systems by design, and a considerable portion of optical theory is developed under this assumption. While some existing deblurring techniques have leveraged rotational symmetry, they are approximate and restricted to a specific subset of radially varying blurs: those due to camera zoom[24,25], the specific case of a parabolic mirror[26] and an approximate scheme only for blurs from a single lens by applying deconvolution to four concentric regions[27–30]. Other work does the same for digital single-lens reflex (DSLR) cameras and also requires red, green and blue (RGB) image channels from a color camera[31]. In contrast, what we propose applies to any rotationally symmetric imaging system, can incorporate more complex PSFs, even if they cannot be theoretically derived, makes no approximations (for example, isoplanatic regions) in the image formation model and can easily extend to other symmetries.

Our ring deconvolution microscopy (RDM) models image formation for rotationally symmetric imaging systems rigorously, allowing for accurate deblurring while remaining practical, both computationally and in terms of calibration. The first step in RDM is a simple, single-shot calibration scheme, in which the system's primary Seidel aberration coefficients are estimated from an image of randomly distributed point sources. These coefficients quantify the severity of spatial variance and provide the necessary system information for the second step, deblurring. We propose two alternative image deblurring algorithms. The first (our main algorithm) is ring deconvolution, which uses a new and rigorous theory for rotationally symmetric imaging to deblur the image at all points in the FoV, with only order $N^3 \log(N)$ ($N$ is the image side length) compute time, as compared to $N^4$ for full spatially varying deblurring. For even faster computation (but without theoretical guarantees), we introduce a neural network-based algorithm called deep ring deconvolution (DeepRD), which constrains learning with physical knowledge provided by the system's Seidel aberration coefficients. If the system is not spatially varying or minimally so, RDM still offers an improvement over standard deconvolution by instead deconvolving with a synthetic PSF generated by the Seidel coefficients.

Although ring deconvolution is specific to systems exhibiting rotational symmetry, our theory can be easily adapted to exploit other forms of symmetry. As an example, we derive an analogous form of deconvolution for situations where the blur varies laterally (along one Cartesian axis), which we term sheet deconvolution. This is the case in light-sheet microscopy, where the light-sheet illumination causes space-varying blur in the direction perpendicular to the imaging plane. We show experimental results for sheet deconvolution on images from a light-sheet microscope.

Our proposed algorithms outperform existing methods, approaching subcellular, isotropic resolution across the FoV. We demonstrate this on four diverse microscope modalities: miniature microscopy, multicolor fluorescence microscopy, multimode fiber micro-endoscopy and light-sheet fluorescence microscopy. Each of these modalities contains different characteristics and imaging mechanisms that are representative of a wide range of imaging systems, thereby forming a comprehensive basis to demonstrate the wide applicability and practical relevance of our methods. Figure 1 provides a summary of the RDM pipeline along with an example result on images of live tardigrades. An open-source implementation of RDM and its extensions can be found in our codebase (https://github.com/apsk14/rdmpy).

## Results

Before we display our experimental results, we briefly outline the RDM pipeline from calibration to deblurring. More details on the RDM implementation can be found in Methods.

### Ring deconvolution microscopy pipeline

The first step in our RDM pipeline is calibration; we measure the system's response to a point source (its PSF). The PSFs of a space-varying system will vary across the FoV and many PSF measurements may be required to fully characterize the system; however, space-varying systems that are rotationally symmetric require fewer measurements for system characterization, which we exploit here. Intuitively, PSFs that are the same distance from the center of the FoV all have the same shape, just rotated at different angles because of the symmetry (Fig. 1a). We call this property of the PSFs linear revolution-invariance (LRI), and denote it mathematically as

$$\tilde{h}(\rho, \phi; r, \theta) = \tilde{h}(\rho, \phi - \theta; r, 0),$$

where $\tilde{h}(\rho, \phi; r, \theta)$ is the (spatially varying) PSF in polar coordinates from a point source at location $(r, \theta)$. Note that the shape of the PSF itself is not necessarily rotationally symmetric. LRI greatly improves complexity of the calibration procedure as we need only measure the PSF at any one point along the circle for each radius $r$ from the optical center.

In practice, directly measuring the PSF at every radius $r$ is still impractical. Instead, under LRI, there is a simple and effective method for simultaneously estimating these PSFs from a single image, without any motion stage required. It works by estimating the Seidel aberration coefficients from an image of point sources randomly scattered across the FoV (Fig. 1b). Seidel coefficients are a natural choice for LRI systems, as Seidel polynomials are explicit functions of the field position and

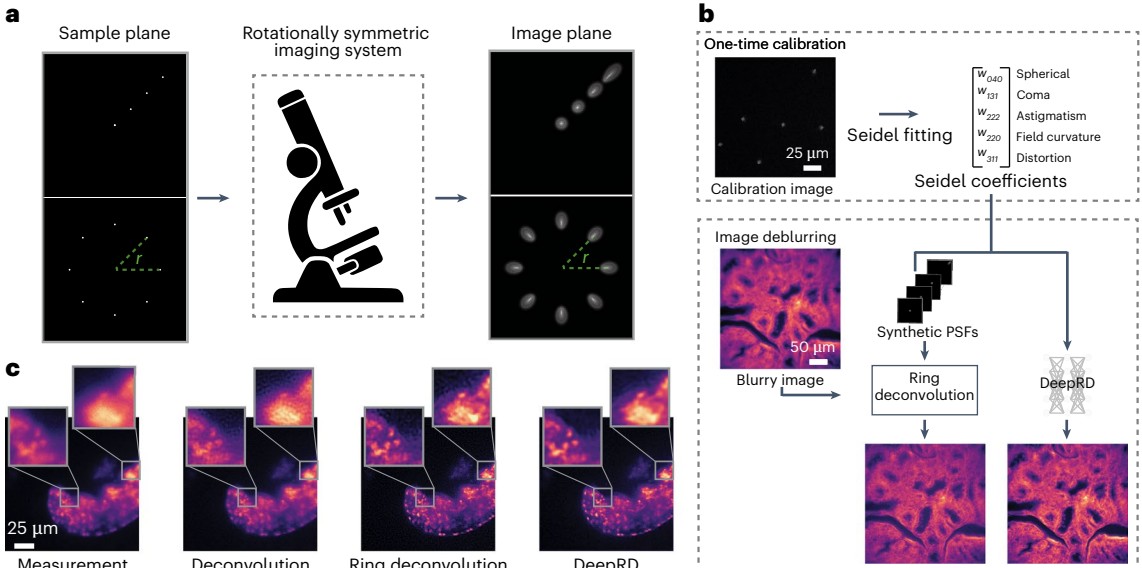

**Fig. 1 | Ring deconvolution microscopy. a**, Point sources at the sample plane (left) are imaged (right) to PSFs with a rotationally symmetric imaging system. The PSFs are LRI; they vary with distance from the center of the FoV (top row), but maintain the same shape at a fixed radius *r*, just revolved around the center (bottom row). **b**, The RDM pipeline. A one-time calibration procedure (top) captures a single image of randomly placed point sources (for example, fluorescent beads) and uses them to fit primary Seidel coefficients. Next, we

either use the Seidel coefficients to generate a radial line of synthetic PSFs, if using ring deconvolution, or we feed the coefficients directly into DeepRD. After calibration, we can deblur images (bottom) using either ring deconvolution or DeepRD. **c**, Experimental deblurring of live tardigrade samples imaged with the UCLA Miniscope[1]. Left to right: measurement, standard deconvolution, ring deconvolution and DeepRD. Ring deconvolution and DeepRD consistently outperform deconvolution.

thus use the same coefficients regardless of the position in the FoV[32]. This makes them a more practical choice than Zernike coefficients, which are different at each field position.

After calibration, the next step is to use the estimated Seidel coefficients to deblur the measured image. LRI systems, much like LSI systems, allow for computationally efficient models. For LSI systems, we leverage two axes of space-invariance to model the blur as a two-dimensional (2D) convolution, and for LRI systems we can leverage the revolution-invariant angular axis, leading to a blur computation that is a sum of one-dimensional (1D) convolutions. We provide a description of the forward and inverse algorithms: ring convolution and ring deconvolution, in Methods. Therein is also included a theoretical proof of ring convolution's exactness under rotational symmetry. We briefly describe ring deconvolution here, along with an alternate method using deep learning, termed DeepRD. These methods form the second step in the RDM pipeline (Fig. 1b).

1. Ring deconvolution. We derive an optimal algorithm for reconstructing the underlying sample from a blurry image given the PSF at each radius of the FoV of a rotationally symmetric imaging system. This is our main algorithm.
2. DeepRD. Although ring deconvolution is considerably faster to compute than a full patch-based spatially varying deblur technique, it may still be relatively slow (on the order of a few minutes) for very large image sizes (for example, beyond 1,024 × 1,024) or video data. Deep learning enables a faster (but approximate) version of ring deconvolution called DeepRD. As input, it takes a blurry image and a list of the five primary Seidel coefficients. DeepRD is trained on a dataset of natural images that are synthetically blurred using ring convolution.

## Experimental results
**Miniature microscopy.** The UCLA Miniscope[1] is a miniature microscope used primarily for neuroimaging in freely behaving animals. Its gradient-index objective, which is required for implantation, causes

spatially varying aberrations thereby limiting its FoV. We demonstrate that RDM can alleviate these limitations. To that end, we first capture a single calibration image of fluorescent beads and fit the five primary Seidel aberration coefficients. We then image a composite U.S. Air Force (USAF) resolution target, rabbit liver tissue and live fluorescence-stained tardigrades. The composite target was made by placing a standard USAF resolution target at nine separate locations in the FoV and then stitching them together, using only the region of each constituent image that contains the highest resolution group.

For each sample, we compare reconstructions from ring deconvolution and DeepRD, as well as standard deconvolution (using the PSF measured at the center of the FoV) and a baseline U-Net (see Fig. 2). Ring deconvolution and DeepRD (having knowledge of the field-varying aberrations via the Seidel coefficients) give the most improvement near the edges and corners of the image. Standard deconvolution produces a noisy, low-contrast result in those regions because of the mismatch of the center PSF with the edge PSFs. We also note that both learning methods (U-Net and DeepRD) offer the best denoising performance, a well-known property of neural networks[33]; however, this comes at the cost of inconsistent performance; both models perform worse on the resolution target than on the other samples. Similar performance is observed for rabbit liver tissue, where our methods reveal features in the corners of the image, including the outlines of membranes that are not clear otherwise. We also capture live videos of tardigrades. We apply deblurring to each frame and display one such frame in the bottom row of Fig. 2; the full videos can be found at https://berkeley.app.box.com/s/d1o1901uv8ehxdf7kzdapyej1c7dt6bi. Ring deconvolution and DeepRD better resolve the small, dot-like features within the tardigrade.

**High-NA multicolor fluorescence microscopy.** Next, we apply RDM to high magnification, high-numerical aperture (NA) microscopy. Such devices are critical to observing biological samples at subcellular resolution; however, as the NA increases, so do field-varying aberrations. RDM offers a pathway to utilize the level of magnification and NA needed for subcellular imaging while maintaining isotropic

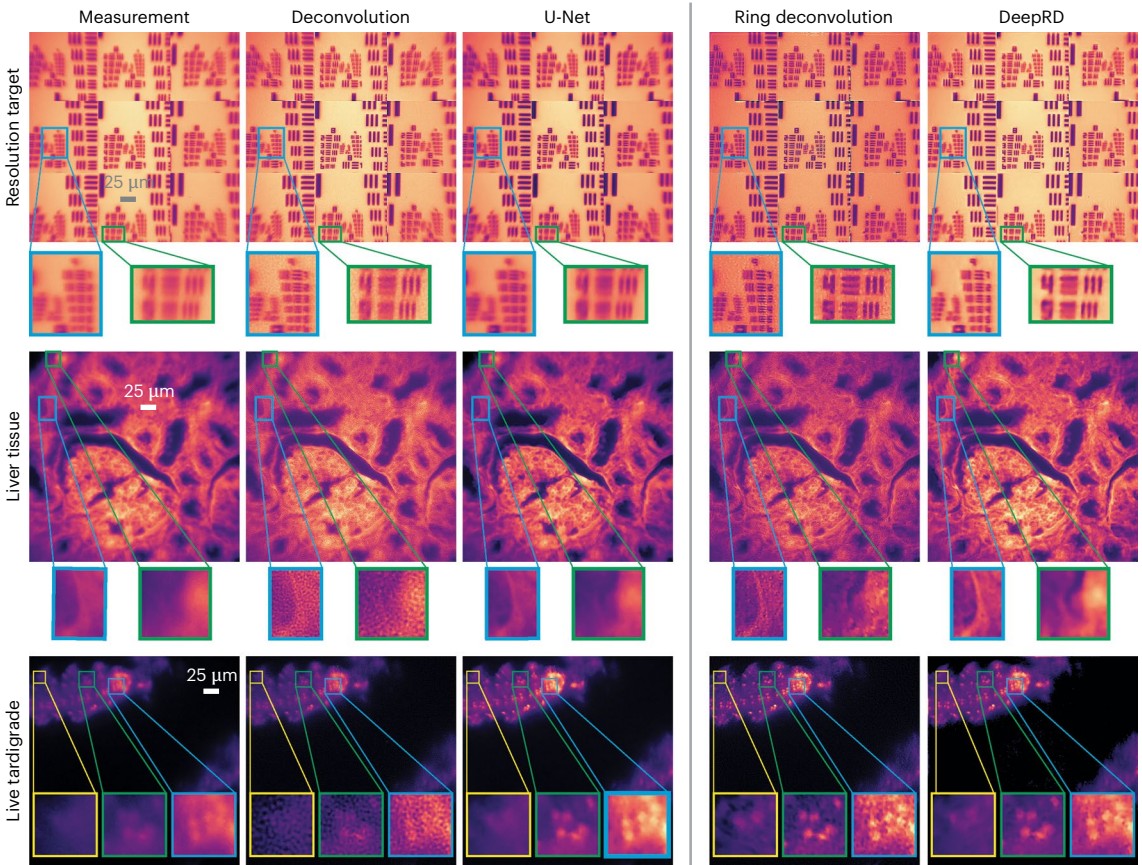

**Fig. 2 | RDM for miniature microscopy.** After calibrating the Miniscope with a single image of fluorescent beads (Fig. 1b), we show results from several deblurring methods for comparison: standard deconvolution (using a single measured PSF), a U-Net trained on our spatially varying blur dataset, ring deconvolution and DeepRD. Deconvolution assumes space-invariance while the remaining methods are designed to handle spatially varying aberrations.

In the first row, ring deconvolution and DeepRD clearly resolve resolution target elements near the edges of the FoV, which are not as well resolved by the two other methods. Zoom-ins show RDM resolves up to element 6 of group 9 (blue inset) and element 5 of group 8 (green inset). Similar results along with corresponding insets are shown for the other samples.

resolution over the entire FoV. Moreover, RDM does this efficiently over multiple fluorescence color channels, allowing for multicolor, subcellular resolution imaging over the entire FoV. To demonstrate this, we image fluorescently labeled actin (green channel) and mitochondria (red channel) of bovine pulmonary artery endothelial (BPAE) cells with a ×100 1.4 NA objective.

We chose to perform a separate calibration for each color channel, with different bead images corresponding to the different emission wavelengths; this strategy allows RDM to additionally correct chromatic aberrations. After calibration, we image BPAE cells and process them with both RDM and standard deconvolution, for comparison. Figure 3 shows that RDM consistently deblurs the raw images throughout the FoV, including the corners of the image, while standard deconvolution becomes low-contrast and noisy near the edges. In both examples, RDM is able to resolve subcellular features in the actin and mitochondria near the edges that are not visible in standard deconvolution. Such capability allows for larger FoVs to be used, lessening the burden of mechanically scanning and stitching together many smaller FoV images when the sample is large.

**Micro-endoscopy through a multimode fiber.** Point-scanning micro-endoscopy through a multimode fiber[15,34] is a powerful technique for deep in vivo imaging at subcellular resolution, with applications in the brain and other sensitive organs, where minimal tissue damage is required; however, due to the extreme constraints imposed

in the design of the fiber, its resolution capabilities degrade rapidly and severely away from the center of the image, resulting in a small usable FoV (top row of Fig. 4). The spatially varying images from such a system have been heuristically deblurred in ref. 15, but RDM, with its rigorous formulation, offers improved performance with far less calibration.

To verify this, we process images of beads and live rat neurons from ref. 15 using RDM, their spatially varying Richardson–Lucy (SVRL) algorithm, and standard deconvolution, for comparison. SVRL is similar to the modal decomposition work in[35,36]. The SVRL method uses fewer PSFs than RDM (30 versus 120), so requires slightly less memory, but requires more calibration images than RDM (441 images versus 1 image). Despite considerably lighter calibration and similar computational complexity, RDM provides an improvement in image quality over SVRL (Fig. 4). In the corners of the bead image, we see that RDM is able to remove the aberration-induced ellipticity of the underlying circular beads and resolve clumps of beads, unlike the other methods. The same holds for the neuron images, where RDM tightens the spread of thin spines far better than the other methods. Additional comparisons on these data are in Extended Data Fig. 2.

In summary, ring deconvolution consistently produces the best reconstructions among the methods we tested. The improvement arises due to the lack of approximations and heuristics in the method's derivation. Moreover, as compared to deep-learning-based methods, ring deconvolution undergoes no learning procedure, and thus does not transmit bias from the training data into future reconstructions.

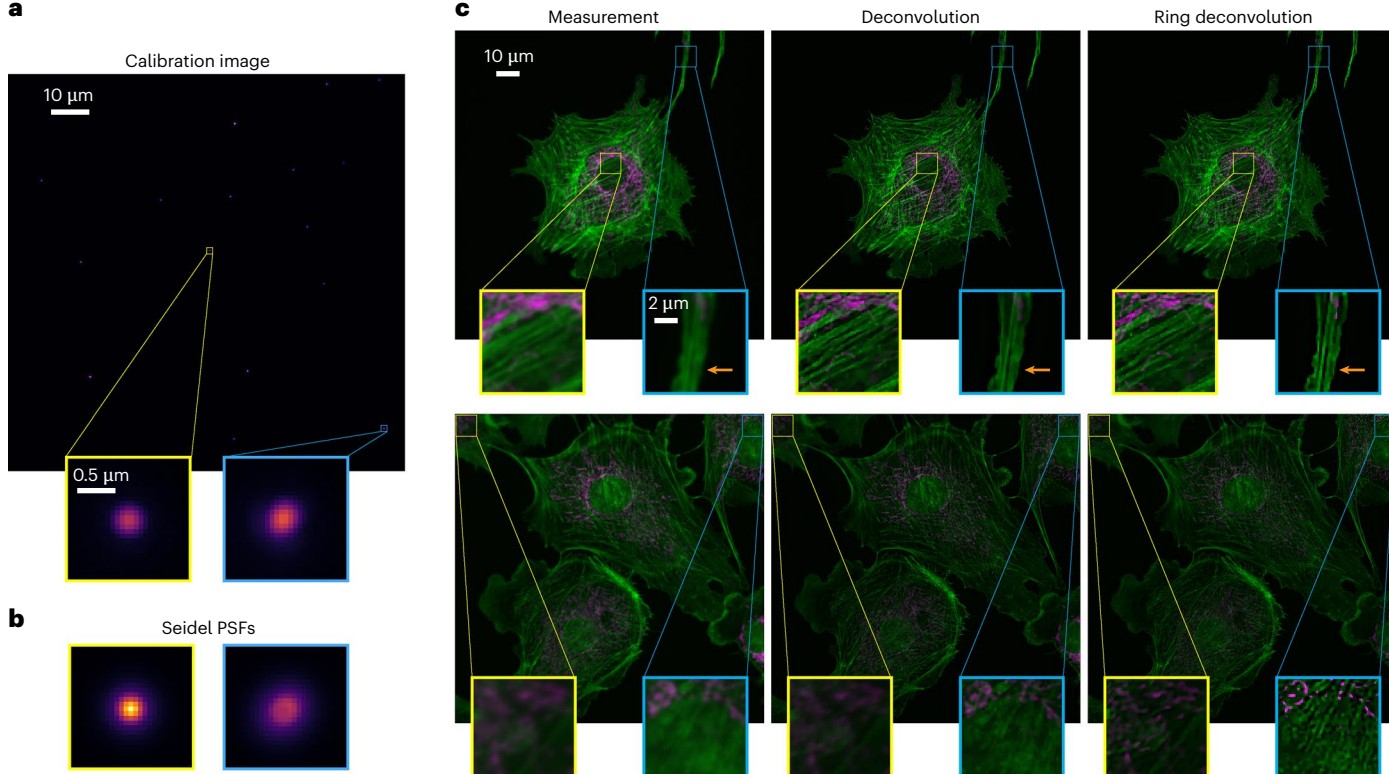

**Fig. 3 | RDM for high-NA, multicolor fluorescence microscopy. a,b,** Fluorescent beads imaged with a ×100 1.4 NA objective (**a**) and corresponding Seidel-fitted PSFs demonstrate spatially varying nature of the system (**b**). **c,** Two representative examples of BPAE cells processed by standard deconvolution and RDM. Deconvolution and RDM perform similarly in the center but RDM is better in the corner, revealing submicron features in the actin (orange arrow). RDM similarly resolves actin filaments and mitochondria where deconvolution does not. A total of six such samples were prepared and imaged with similar results.

DeepRD performs similarly to ring deconvolution, but is faster and less consistent. Both perform better than the U-Net and standard deconvolution.

## Simulation results

To quantify the performance of our methods, we conducted a series of simulations in which we have access to the unblurred ground-truth image. These simulations are performed with Gaussian noise and are repeated with Poisson noise (Supplementary Fig. 2).

Our first step is to quantify the forward model: ring convolution. When a system is space-invariant, the forward model is a simple convolution operation. For space-varying systems, however, we must account for the changes in the PSF across the FoV. The brute-force approach for doing so would superimpose weighted PSFs at each pixel in the image to compute the 'true blur', at the cost of long compute times; however, when the system is rotationally symmetric (varies only radially), ring convolution is an equivalent operation to the brute-force method, but runs much quicker (less than a second) even for image sizes upward of 512 × 512. To verify, we blur each test image using spatially varying PSFs rendered from a randomly chosen set of Seidel coefficients. We treat the true blur as ground truth and compare the error maps for both standard convolution and our ring convolution. As expected, standard convolution results in errors near the edges of the FoV, whereas ring convolution produces accurate blur across the entire image (Fig. 5a). In Fig. 5b we see that the error for standard convolution increases approximately linearly with this aberration magnitude (the norm of Seidel coefficients). In contrast, ring convolution continues to produce an accurate blur, independent of the strength of the aberrations. In Fig. 5c, we compare compute times for these forward models, showing that our ring convolution method is nearly four orders of magnitude faster than the other exact method (true blur) for a megapixel-sized image.

Next, we verify our Seidel fitting method by quantifying the error in our estimated Seidel coefficients that were fitted from a single noisy image of randomly scattered point sources. As detailed in Methods, the Seidel fitting procedure involves searching for the set of five primary Seidel coefficients that best fit the measured PSFs at their given positions in the calibration image. We simulate this process many times by randomly generating sets of Seidel coefficients, using them to produce a calibration image with additive Gaussian noise, and then estimating those coefficients only using the noisy calibration image (Fig. 5d). We plotted the error between the fitted and true coefficients in Fig. 5e. Due to nonconvexity of the fitting problem, we see that not every case produces errors that converge to 0; however, two things provide assurance: (1) the median convergence approaches 0, meaning that a majority of optimizations will produce the optimal solution; and (2) even the runs that do not converge to the global minimum produce PSFs, which are close enough to the true ones to provide good quality ring deconvolution. Finally, as the Seidel fitting procedure acts as a PSF denoiser, we test the fit with varying amounts of noise (up to −20 dB signal-to-noise ratio (SNR)) and find that the fit is still accurate even with severe additive noise (Fig. 5f). Additional simulations for higher-order Seidel coefficients can be found in Supplementary Fig. 3.

After verifying that our forward model and Seidel fitting methods perform well, we use them as part of an inverse problem to deblur images and quantify the performance. We compare our methods (ring deconvolution and DeepRD) with standard deconvolution and the baseline U-Net. The learning methods are trained with images from the Content-Aware Image Restoration (CARE) and Div2k datasets[21,37], after synthetically blurring them with space-varying PSFs rendered from a random set of Seidel coefficients. It is computationally infeasible to generate the entire training set using the true blur technique, so instead we use our ring convolution to generate the blurred images for

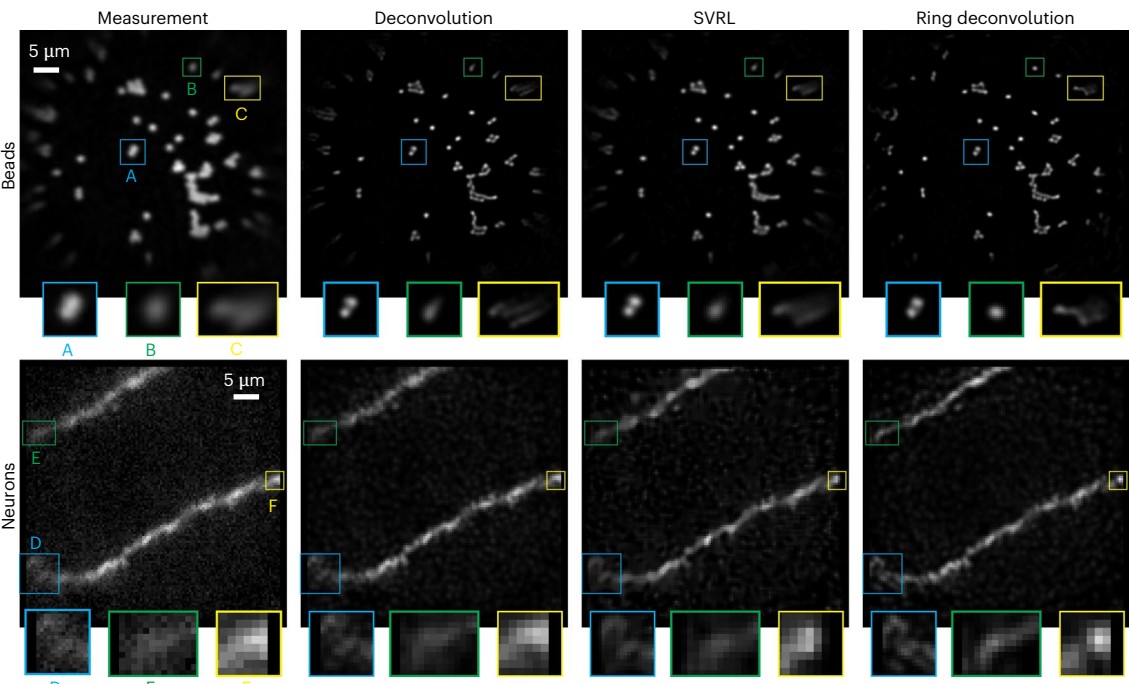

**Fig. 4 | RDM for point-scanning micro-endoscopy through a multimode fiber.**
Comparison of deconvolution, SVRL and RDM on images taken from a point-
scanning multimode fiber micro-endoscope. The raw images and the SVRL
results are obtained directly from ref. 15. Top: sample with 1-μm beads;
all methods resolve the circularly shaped beads in the center (blue inset).
Away from the center, however, deconvolution and SVRL fail to completely
remove the ellipticity of the bead but RDM does (green inset). Moreover, in the
corner only RDM can resolve bead clusters into their component beads
(yellow inset). Bottom: live Wistar rat neuron; RDM can clearly resolve structures
(blue inset), sharpen the neuron spine (green inset) and reveal a point-like
structure (yellow inset) near the edges, whereas the other methods cannot.

training and use the true blur for the test set. The results of each method
on one representative test image are shown in Fig. 5g,h, with the peak
SNR (PSNR) (in dB) listed above. Both DeepRD and ring deconvolution
deblur better near the edges and corners of the image, where the PSF
deviates the most from the center PSF. Despite using ring convolution
for the training set and the true blur for the test set, neither of the net-
works (U-Net and DeepRD) show signs of model mismatch.

In Fig. 5i, we plot the average accuracy (PSNR) versus runtime
across the entire test dataset (28 images), with the size of the circle
representing the number of parameters needed (memory footprint)
for each method. Ring deconvolution provides the best reconstruction,
but it is also the largest and slowest of the methods tested, taking about
60 s for a full image; however, if we were to try to deblur the image rig-
orously without using RDM, it would take hundreds of hours, despite
being theoretically equivalent to ring deconvolution. Thus, ring decon-
volution allows for relatively fast, accurate deblurring where it was once
infeasible. DeepRD performs nearly as well as ring deconvolution and
has the fewest parameters needed of all the space-varying techniques,
allowing it to be fast and memory efficient. DeepRD is almost three
orders of magnitude faster than RDM. The baseline U-Net and standard
deconvolution PSNR values are considerably worse.

### Sheet deconvolution: extension to lateral symmetry

Revolution-invariance is not the only form of symmetry found in practi-
cal, spatially varying systems. For example, light-sheet fluorescence
microscopy (LSFM) exhibits lateral symmetry, in which light-sheet
illumination causes spatial variance, but only along one axis (Fig. 6a).
The thinnest section, or waist, of the light-sheet is focused in the center
of the FoV and becomes thicker as a function of the distance from the
center. As the light-sheet is constant along the axis orthogonal to the
focusing direction, this variance only occurs along one dimension.
The resulting PSF grows in axial extent as it moves along one of the
lateral dimensions, but stays constant in the other lateral dimension.

Here we derive sheet convolution, an exact forward model for
LSFM (assuming a space-invariant objective lens), which leverages
symmetry by convolving over the two dimensions that are spatially
invariant, while integrating over the one that does vary. As a result,
sheet convolution enjoys a similar improvement in computational
performance as ring convolution; instead of the $O(N^6)$ scaling of the
general three-dimensional spatially varying forward model, it achieves
a $O(N^4 log(N))$ scaling. This improvement enables rigorous, spatially
varying LSFM deconvolution where it once was computationally infea-
sible. All that remains is to obtain the three-dimensional (3D) PSF at
each point along the light-sheet focusing axis. This can be carried out in
multiple ways using only a single calibration volume; the experimental
PSFs from this volume are either interpolated at the missing locations
or are used to fit the parameters of a PSF model. Details along with the
derivation of sheet convolution/deconvolution are in Methods and
simulations are in Supplementary Fig. 1.

To combat the inherent spatial variance of LSFM, practitioners
typically use only the thin lateral slab of the image that lies within the
light-sheet waist, then shift the sample repeatedly to acquire the full
FoV[38]. The final image is then stitched together from multiple tiled
acquisitions; however, by using our sheet deconvolution approach,
the parts of each acquisition that lie outside of the waist can be recov-
ered computationally, thereby expanding the usable FoV of the LSFM
system and reducing the number of acquisitions needed for an object
with a large lateral extent. While there are techniques, like axially swept
LSFM[39], which speed up the process in hardware, a purely computa-
tional solution is desirable for its ease of use, reduced irradiation and
flexibility.

Figure 6 shows a demonstration of sheet deconvolution on LSFM
data. For the experiment, a 3D stack of randomly scattered fluorescent
nanospheres in agarose gel was imaged and used to obtain PSFs at
each position along the spatially varying axis (Methods). Then, using
these PSFs, sheet deconvolution is applied to samples of two types:

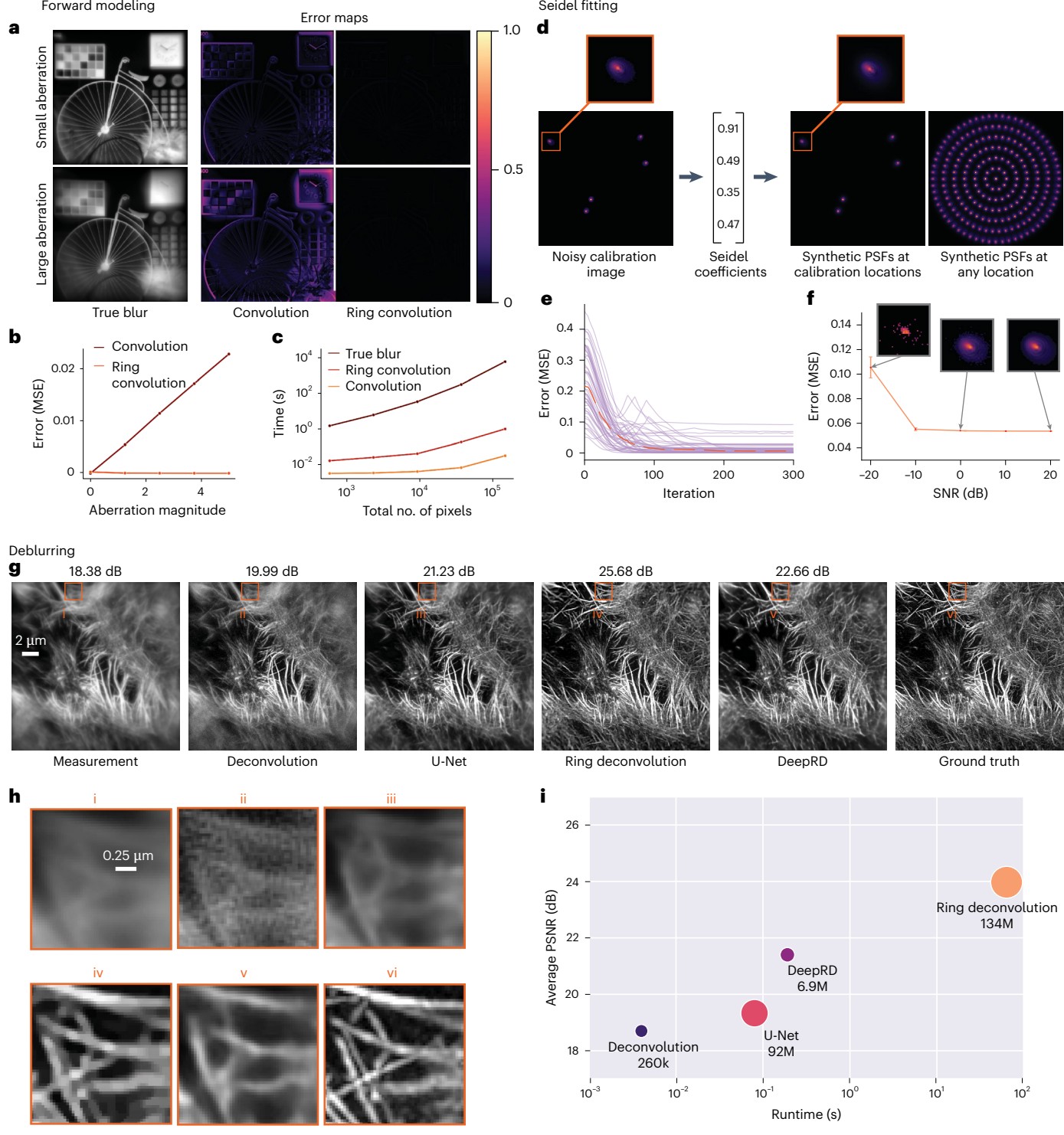

**Fig. 5 | Simulations to quantify RDM performance. a**, Error maps showing the absolute difference between ring convolution/standard convolution and the 'true blur' (produced by manually superimposing every PSF at every pixel). When off-axis aberrations are small (top row), both forward models are accurate. When aberrations are large (bottom row), convolution becomes noticeably worse, yet ring convolution remains accurate. **b**, Forward model mean-squared error (MSE) as a function of off-axis aberration magnitude (for the image above). **c**, Runtime of each method as a function of size of the image (in pixels), averaged over $n = 50$ trials. **d**, Seidel coefficients are fit to a noisy image of randomly placed PSFs and then used to generate PSFs at any location. **e**, Seidel fit error at every iteration of the fitting algorithm. Each purple line is one of $n = 50$ trials, each with

a different, random set of underlying Seidel coefficients. The red dashed line is the per-iteration median. **f**, Average MSE of the fitted Seidel coefficients plotted against the SNR of the calibration image (with standard deviation shown by error bars) over $n = 50$ random trials. Some example calibration PSFs are shown. **g**, Deblurring results on noisy images from the CARE dataset, with PSNR values above each method. **h**, Zoom-ins of an off-axis patch in each deblurred image; ring deconvolution and DeepRD have the highest quality. **i**, Average accuracy in PSNR versus runtime of each method over $n = 28$ true blurred images using unseen coefficients. The number of model parameters is written below each circle and determines its size.

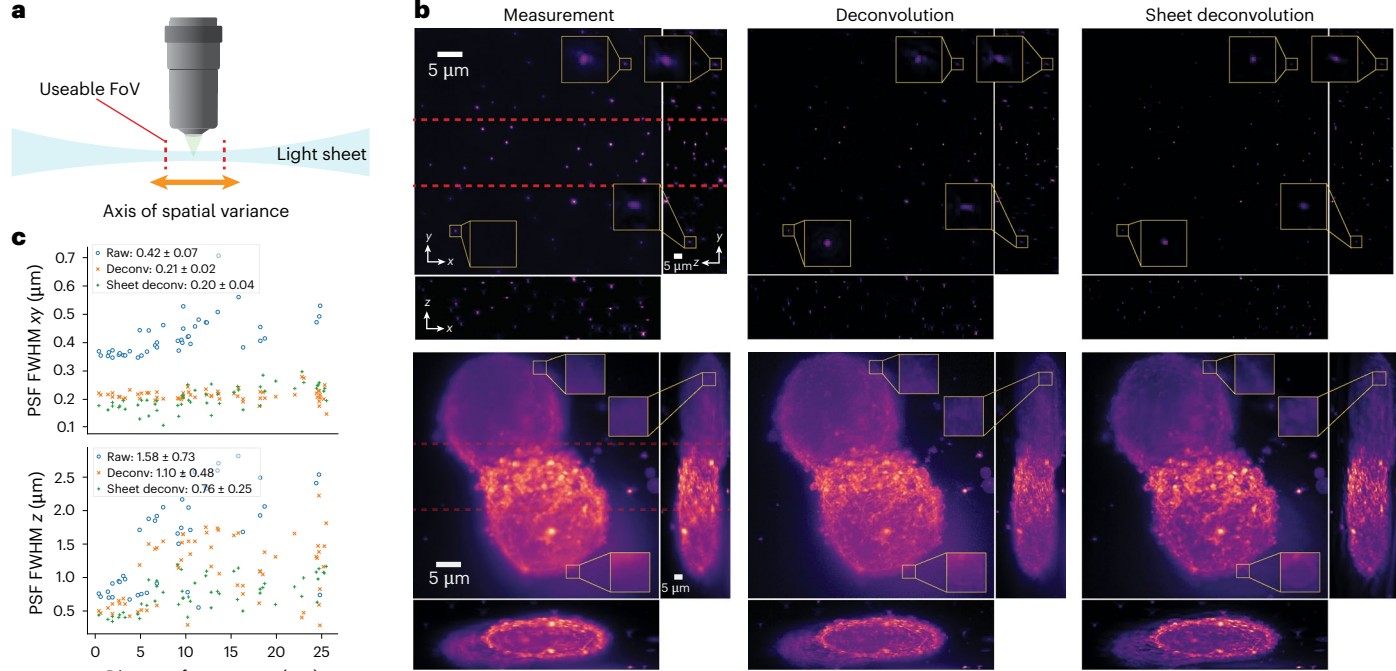

**Fig. 6 | Sheet deconvolution exploits lateral symmetry for light-sheet microscopy.** We run sheet deconvolution on samples imaged by a light-sheet fluorescence microscope with a detection NA of 1.1 and a light-sheet NA of 0.31. **a**, The light-sheet microscope illumination spreads along its propagation axis (orange arrow) such that only a center strip in the FoV (marked 'usable FoV') is optically sectioned and hence well resolved. **b**, Results comparing standard 3D deconvolution and sheet deconvolution on fluorescent beads (top row) and

SU8686 cells (bottom row), with maximum-intensity projections along each axis. Unlike standard deconvolution, sheet deconvolution resolves small features nearly as well at the edges of the FoV as in the center (insets). **c**, Plots of lateral (top) and axial (bottom) FWHM of reconstructed beads (from **b**) as a function of their distance from the center along the spatially varying axis. The plot legends show the average FWHM plus or minus the s.d.

fluorescent beads and SU8686 cells. Unlike standard deconvolution, sheet deconvolution is able to better resolve features that do not lie in the light-sheet waist and create a more homogeneous axial resolution across the entire FoV than standard deconvolution. We quantified the resolution improvement by measuring the lateral and axial full-width half-maximum (FWHM) of fluorescent beads across the entire FoV after applying the two deconvolution methods. As expected, the lateral FWHM does not change substantially as the PSF spatial variance occurs primarily in $z$, but the axial FWHM values from sheet deconvolution are on average 300 nm smaller than those from standard deconvolution with 200 nm less s.d. This increase in resolution outside of the beam waist can reveal subcellular-scale features that were previously only accessible by shifting the sample. In this experiment, sheet deconvolution ran on a volume size of 512 × 512 × 160 in about 7 min, which is orders of magnitude faster than solving a spatially varying deblurring method without leveraging symmetry. With sheet deconvolution, users of LSFM can speed up the capture of large samples while still getting high resolution across the entire FoV.

### Space-invariant systems

We have demonstrated our methods for rotational and lateral symmetry, and expect that the extension to other symmetries would be analogous. In addition, elements of RDM may find use even when the system is not space-varying or when calibration data are not available. If aberrations are not space-variant, the first part of the RDM pipeline can still provide value by estimating the spherical aberration coefficient from a calibration image via Seidel fitting. With this coefficient, we can generate a synthetic center PSF and perform deconvolution. We call this procedure Seidel deconvolution, and find that it essentially denoises the PSF measurement as it finds the closest synthetic PSF to the measured one. Additionally, this single coefficient can be jointly

estimated with the deconvolved image with no calibration, providing a technique for blind deconvolution (details are in Methods).

In the results shown in Extended Data Fig. 1, Seidel deconvolution resolves smaller features and gives a cleaner reconstructed image than standard deconvolution with the measured PSF. The blind deconvolution result is similar to that of Seidel deconvolution, though cannot resolve the smallest features and has overly high contrast due to noise. Future work extending this idea to blind, spatially varying deblurring could use DeepRD to iteratively search over the space of deblurring networks and choose the network with the sharpest reconstruction.

We note that while existing work has fit a variety of parametric models to the experimentally measured PSF, including a 2D Gaussian distribution[40], Gaussian mixture model[41], Zernike basis[42–47], spherical aberration diffraction model[48] and Seidel ray model[49,50], fitting Seidel polynomials in the pupil function is new.

## Discussion

In summary, we developed a pipeline for image deblurring in rotationally symmetric systems, called RDM, which encompasses both an analytically derived deblurring technique, ring deconvolution and its fast alternative, DeepRD. Like standard deconvolution microscopy, our methods only require a single calibration image; however, they offer space-varying aberration correction. We support RDM with a new theory of imaging under rotational symmetry, which we call LRI, and an implementation of the LRI forward model (ring convolution). For RDM calibration we also develop a procedure for fitting Seidel aberration coefficients from a single calibration image of randomly placed point sources. To show generality of our ideas, we further derive, implement and test an analogous method that exploits linear (instead of radial) symmetry, for applications in light-sheet microscopy.

We verify the accuracy of our deblurring methods in both simulation and experimentally over four diverse microscope modalities.

We hope that RDM will ultimately replace deconvolution microscopy as standard practice in widespread applications from biology to astronomy. We believe that RDM will find most use in systems that approach optical extremes such as miniature microscopes or large FoV systems, but may also empower optical designers to simplify hardware knowing they have the ability to better correct for aberrations digitally. RDM is well suited to dynamic conditions in which the system or sample is changing, as long as the system calibration can be updated accordingly. For simpler cases that can be directly modeled, such as the time-varying deformation of an imaging fiber, it should be possible to update the initial calibrated PSFs with a theoretical model[51,52].

We intend RDM to be a living, breathing tool with constant improvements to its speed, accuracy and artifacts. Already we have seen a 20-fold decrease in ring deconvolution's runtime in preliminary experiments by parallelizing it over multiple graphical processing units (GPUs) using the novel Chromatix optical framework[53]. Moreover, the constant improvement in deep-learning architecture, including better conditional models[54], can also improve the performance of DeepRD in future. We plan to continually update our codebase with these improvements.

## Online content

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

## Methods

### Ring convolution and ring deconvolution

We begin with a primer on notation. Let $g(u, v)$ describe the object's intensity at $(u, v)$ and $h(x, y; u, v)$ describe the space-varying PSF; the intensity at $(x, y)$ of the PSF generated by a point source at $(u, v)$. We further use the notation $\tilde{g}$ to denote the transformation of $g$ to polar coordinates. Then the final image intensity $f(x, y)$ of a linear optical system is formed by the superposition integral[55]:

$$f(x, y) = \int \int g(u, v) h(x, y; u, v) du dv. \tag{1}$$

This equation is the system forward model for a linear space-varying system; it describes the image as a function of the object and PSFs at different locations in the FoV.

Standard image deconvolution approximates the system as LSI, which means the PSF is the same at all positions in the FoV, $h(x, y; u, v) = h(x - u, y - v; 0, 0)$. This simplifies the forward model in equation (1) by reducing it to a convolution with a single PSF. This greatly reduces the computation for forward and inverse problems, but at the cost of being inaccurate for space-varying aberrations.

In this paper, we incorporate radially varying aberrations analytically into the forward model to provide a middle ground between purely space-invariant and completely space-varying systems. The assumption is that the system is LRI (its physical configuration is symmetric about the optical axis; Fig. 1a). This is true of many typical optical imaging systems. Our core observation is that all LRI optical systems satisfy

$$\tilde{h}(\rho, \phi; r, \theta) = \tilde{h}(\rho, \phi - \theta; r, 0). \tag{2}$$

Under this assumption, the object intensity can be written as what we call a ring convolution, denoted by $f \triangleq g \circledcirc h$. By substituting equation (2) into equation (1) with $x = \rho \cos\phi$ and $y = \rho \sin\phi$, we get

$$f(x, y) = (g \circledcirc h)(x, y) = \int_0^\infty r(\tilde{g} *_\theta \tilde{h}) \left( \sqrt{x^2 + y^2}, \tan^{-1}(y/x); r, 0 \right) dr, \tag{3}$$

where the $*_\theta$ operator indicates a 1D convolution over the $\theta$ dimension. The $r$ arises in the deconvolution as we are integrating over object space $(u, v)$ in polar coordinates $(r, \theta)$. This ring-wise computation, wherein points at different radii are filtered heterogeneously, is consistent with the underlying intuition in LRI: the blur varies radially. Our first main result allows for an efficient, fast Fourier transform (FFT)-based version of ring convolution.

**Theorem 1.** Ring convolution theorem. Under LRI, where $\mathcal{F}_\Theta$ is a 1D Fourier transform over $\theta$,

$$\tilde{f}(\rho, \phi) = \mathcal{F}_\Theta^{-1} \left\{ \int r \mathcal{F}_\Theta \{ \tilde{g}(r, \theta) \} \mathcal{F}_\Theta \{ \tilde{h}(\rho, \theta; r) \} dr \right\}(\phi).$$

**Proof.** As the given system is LRI, equation (3) holds. Substituting $\rho = \sqrt{x^2 + y^2}$ and $\phi = \tan^{-1}(y/x)$, we can rewrite equation (3) as

$$\tilde{f}(\rho, \phi) = \int_0^\infty r(\tilde{g} *_\theta \tilde{h})(\rho, \phi; r, 0) dr.$$

Applying the Fourier convolution theorem to the 1D convolution on the right-hand side yields

$$\tilde{f}(\rho, \phi) = \int r \mathcal{F}_\Theta^{-1} \left\{ \mathcal{F}_\Theta \{ \tilde{g}(r, \theta) \} \mathcal{F}_\Theta \{ \tilde{h}(\rho, \theta; r) \} \right\}(\phi) dr,$$

where $\mathcal{F}_\Theta$ is the 1D Fourier transform over $\theta$. By Fubini's theorem, we pull the inverse Fourier transform outside of the integral, which gives

$$\tilde{f}(\rho, \phi) = \mathcal{F}_\Theta^{-1} \left\{ \int r \mathcal{F}_\Theta \{ \tilde{g}(r, \theta) \} \mathcal{F}_\Theta \{ \tilde{h}(\rho, \theta; r) \} dr \right\}(\phi).$$

There is an efficient and convex formulation for computing the inverse of ring convolution, namely ring deconvolution:

$$\hat{g} = \underset{\tilde{g}}{\arg\min} \| f - \tilde{g} \circledcirc h \|_2^2. \tag{4}$$

This problem can be solved via an iterative least squares solver using Algorithm 1 as a substep. A Fourier interpretation of ring convolution is provided in Supplementary Fig. 5. While the above results are all rigorous, the discrete time implementations of them have small, but nonzero errors due to discretization. For example, the polar transformation in Algorithm 1 requires a small amount of interpolation. As is the case with standard deconvolution, there are conditions for which ring convolution is not fully invertible and consequently ring deconvolution will not recover the sample exactly. The diffraction limit, for example, manifests in PSFs whose frequency spectrum is bandlimited, rendering frequencies in the sample that are past the bandlimit irrecoverable. This can also happen if certain frequencies in the system's transfer function are below the noise floor. In such cases ring deconvolution, because it is convex, will return an estimate of the sample that is closest in $l2$ norm to the true sample. Regularization can improve this result even further by leveraging previous knowledge about the sample in question. Visualizations of these algorithms can be found in the Supplementary Information.

**Algorithm 1.** Ring convolution

**Input:** $N \times N$ pixel image $g$; PSFs along one radial line $h^{(j)}, j = 1, \dots, K$; corresponding distances $r_j, j = 1, \dots, K$ of each PSF from the center.

**Output:** LRI blurred image $f$

1: $\tilde{g} \leftarrow \text{polarTransform}(g)$ ▷ polar dimensions are $M \times K$, angle by radius
2: $\tilde{f} \leftarrow \text{zeros}(M \times K)$ ▷ initialize the output in polar form as an all zero matrix
3: **for** $j = 1, \dots, K$ **do**
4: $\quad \tilde{h}^{(j)} \leftarrow \text{polarTransform}(h^{(j)})$
5: $\quad$ **for** $i = 1, \dots, K$ **do**
6: $\quad\quad \tilde{f}_{:,i} \leftarrow \tilde{f}_{:,i} + \text{iFFT}\{r_j \text{FFT}\{\tilde{g}_{:,j}\}\text{FFT}\{\tilde{h}_{:,i}^{(j)}\}\}$ ▷ compute polar output ring by ring, FFT is 1D

7: $f \leftarrow \text{inversePolarTransform}(\tilde{f})$

### Fitting Seidel coefficients to PSFs

Ring (de)convolution algorithms require $h$, the collection of PSFs along one radial line in the FoV. Fortunately, there is a convenient and compact alternative to measuring these manually. The Seidel aberration coefficients[32,56] are a polynomial basis that can represent any rotationally symmetric system. We mathematically describe the form of these aberrations in the Supplementary Information.

The estimation procedure for estimating the Seidel coefficients involves fitting them to a single, sparse image of a few randomly scattered point sources (for example fluorophores on a microscope slide). Such an image is usually easier to obtain than an image of an isolated point source in the center of the FoV. The presence of off-axis PSFs in the calibration image provides information about all aberration coefficients. Though it may be possible to fit these coefficients from a single off-axis PSF, we find that a few, randomly placed PSFs provide a robust fit. The locations of the point sources are not known a priori and are estimated via local peak detection. The optical center of the system is also needed to properly localize the PSFs; this can be found heuristically or by using common optical center finding algorithms such as ref. 57.

Let $r_1, \dots, r_J$ be the object-plane radii of the $J$ points in the calibration image. We then find the primary Seidel coefficients $\hat{\omega}$ whose generated PSFs best match the measured PSFs. Once again, this searching procedure is succinctly stated as an optimization problem,

$$\hat{\omega} = \underset{\bar{\omega}}{\arg\min} \sum_{j=1}^{J} \| h^{(j)} - \mathcal{F}^{-1}\{P(\bar{\omega})^{(j)}\} \|_2^2, \tag{5}$$

where $P(\bar{\omega})^{(j)}$ is the pupil function with Seidel coefficients $\bar{w}$ from a point source at distance $r_j$ from the center of the FoV.

It has been shown that for LRI systems, the optimal fit $\hat{w}$ achieves a diminishing error[32]. Furthermore, the five primary Seidel coefficients index the dominant aberrations present in practical imaging systems: sphere, coma, astigmatism, field curvature and distortion. For more complex aberrations it is possible to add higher-order Seidel coefficients to the fit (Supplementary Information); however, our experiments demonstrate that the five primary coefficients suffice to characterize practical, spatially varying LRI systems. In practice, we fit these five coefficients via the ADAM optimizer[58] and obtain reasonable local minima even though the problem is nonconvex. We note that this procedure is optimal in the maximum likelihood sense[59]. Armed with the estimated Seidel coefficients, we can generate synthetic PSFs at any radius. Note that Seidel coefficients are easier to estimate than Zernike coefficients in a similar fashion, which are different at each field position[42,43].

### Deep ring deconvolution

DeepRD accepts a two-part input, a blurred image and its corresponding primary Seidel coefficients. DeepRD is designed to incorporate these Seidel coefficients into the deblurring process in a parameter-efficient and interpretable manner. To that end, we propose a neural network architecture inspired by the physical LRI image formation model. The first key design element is to use a modified Hypernetwork[60], a network that predicts the weights of another, task-specific 'primary' network. In DeepRD, an multilayer perceptron (MLP)-based hypernetwork takes in Seidel coefficients and produces a deep deblurring network that specifically works for the given coefficients. Our second key design element is the use of ring-wise convolution kernels. Specifically, the hypernetwork produces CNN kernels for each radius which the primary network applies ring by ring. This replicates the revolution-invariance assumption central to ring deconvolution and eases the space-invariant constraint of typical convolutional kernels. Together, this design enables a neural network that is a fraction the size of a conventional U-Net with improved performance and generalization.

To produce a training dataset for DeepRD, we synthetically generate blurred input images from the Div2k and CARE fluorescence microscopy datasets using ring convolution with randomly sampled sets of Seidel coefficients. We must sample coefficients which adequately cover the attributes of realistic imaging systems. Each coefficient, which is in units of waves, is drawn from a uniform distribution and a noise-perturbed grid in the range 0–3 waves (we empirically find this range to cover the aberrations of systems ranging from perfect to highly aberrated). Note that without ring convolution, such a dataset would be prohibitively slow to produce. With that in mind, we will release both an implementation of ring convolution as well as our dataset. Each model is first pretrained on 80,500 data points from the Div2k dataset (800 base images blurred with 100 different Seidel coefficients) and then fine-tuned on a small 8,400 data point subset of the CARE dataset (24 images and 350 different Seidel coefficients). After training, each method is tested on a test set of 28 unseen images from the CARE dataset, which are blurred using the true blur method and noised with additive Gaussian noise.

We find that this physically grounded synthetic dataset generation is effective in training models that generalize to real-world evaluation. As further evidence of generalization, DeepRD seems to disentangle the effects of each aberration coefficient. An exploration of this interpretability is found in the Supplementary Information.

### Sheet convolution, deconvolution and the LSFM PSF model

Here we derive sheet convolution and deconvolution mathematically. The derivations are roughly similar to those for ring convolution, but with a different symmetry. Because of the similarities, our exposition here is more terse.

### Sheet convolution and deconvolution.

LSFM is a 3D imaging modality and therefore the object and PSF will be a function of three space variables. Let $g(u, v, t)$ describe the object intensity at location $(u, v, t)$, and $h(x, y, z; u, v, t)$ describe the space-varying PSF due to Gaussian light-sheet illumination focused along the $u$ dimension (the choice of $u$ as the first spatial dimension is arbitrary). The superposition integral for a linear optical system can then be written as

$$f(x,y,z) = \int \int \int g(u,v,t)h(x,y,z;u,v,t)dudvdt,$$

analogously to equation (1). As before, we will incorporate the symmetry of LSFM to simplify the above display. The symmetry assumption states that the imaging optics are space-invariant, but the PSF varies in the $u$ direction due to the beam profile. Recall also that the total PSF is the product of the imaging PSF with the illumination PSF, which is varying. The following equation encodes these assumptions:

$$h(x, y, z; u, v, t) = h(x, y - v, z - t; u, 0, 0).$$

As before, plugging this into the linear forward model gives us sheet convolution:

$$f(x,y,z) = \int\int\int g(u,v,t)h(x,y-v,z-t;u,0,0)dudvdt$$
$$= \int(g *_{v,t} h)(x,y,z;u,0,0)du,$$

where $*_{v,t}$ represents a 2D convolution along the $v$ and $t$ axes. From the equation we see that the image is the integral of 2D convolutions of $y - z$ sheets of the object with $y - z$ sheets of the PSFs. To compute this integral, we only need to know the PSF at all values of $u$, and the $v$ and $t$ dimensions do not matter.

With a forward model in hand, sheet deconvolution can be solved the same way as ring deconvolution: iterative least squares (equation (4)). The discretization and computational implementation of these algorithms mimic those of ring convolution and can be found in our codebase.

### LSFM PSF model.

The remaining component of LSFM deblurring strategy is to obtain the necessary set of 3D PSFs along one dimension. Similar to our earlier strategy for calibrating ring convolution/deconvolution, we can do so by imaging a single calibration volume containing randomly located, sparsely distributed point sources (for example, fluorescent beads embedded in agarose). The resulting image stack will contain a random set of PSFs at different $u$ locations.

The simplest option is to directly estimate the PSFs at the missing $u$ locations by taking the convex combination of the two closest measured PSFs from the calibration stack. We use this strategy for the bead experiment in Fig. 6.

The second option is more akin to our Seidel fitting procedure; it involves parametrization of the spatially varying PSFs with a unified, differentiable model. In this case we develop a modified version of popular Gibson–Lanni 3D PSF model for LSFM[61]. We will focus on our modifications of the model; further details about the Gibson–Lanni model and its variants are ubiquitous in the literature[61,62]. Given a vector $p$ of system characteristics (for example, sample refractive index) the Gibson–Lanni model calculates the optical path difference between the ideal and experimental imaging systems. Integrating over this optical path difference gives the system's 3D PSF. Our LSFM PSF model takes this PSF and truncates its $z$ extent according to the light-sheet illumination thickness, thereby creating a spatially varying PSF in $u$. Formally, let the Gibson–Lanni PSF be $h_p$, then our LSFM PSF

$$h(x,y,z;u,v,t) = h_p(x-u,y-v,z-t)\frac{1}{\sigma(u)\sqrt{2\pi}}e^{-\frac{1}{2}\left(\frac{z-t}{\sigma(u)}\right)^2},$$

where

$$\sigma(u) = \alpha\sqrt{1 + (\beta u)^2}.$$

The above equations arise from modeling the light-sheet as a Gaussian beam along $u$; its spread $\sigma(u)$ changes hyperbolically along the focus direction and its profile at a given $u$ is a Gaussian with variance $\sigma(u)^2$. We have two control knobs for it, $\alpha$, which controls the $z$ spread of the PSF at the thinnest section or waist, and $\beta$, which controls the rate that the spread increases as a function $u$. Given the calibration stack, we can optimize $\alpha$ and $\beta$ such that our PSF model produces PSFs close to the experimental ones. Since the PSF is a differentiable function of $(\alpha, \beta)$, they can be solved for using gradient-based iterative optimization, just like for equation (5). It is also possible to incorporate elements of $p$ into this optimization if they are not known a priori. This model is used to calibrate sheet deconvolution on the cell sample in Fig. 6.

To deploy the above models experimentally, one must ensure that the point sources are sufficiently sparse such that the PSFs are mostly nonoverlapping. The exact point source locations are not important and can be estimated. Noisy PSFs are best handled with the Gibson–Lanni fitting method, which uses synthetic PSFs but still fits the measured PSFs well (Supplementary Fig. 1). The interpolation method is more sensitive to noise but can be denoised effectively with thresholding and median filtering. The fitting procedures are detailed in our codebase and follow the same general structure: first take a calibration stack of randomly scattered point sources, estimate the locations of the PSFs using local peak finding algorithms, produce generated PSFs at those locations and use the error between the generated and measured PSF to update the PSF model. In the case of interpolation, the last step is replaced by linearly interpolating the two closest measured PSFs to the desired location to create the generated PSF there.

## Blind deblurring

Our version of blind deconvolution also takes advantage of the Seidel coefficients. Given just a blurry image, we start by randomly picking a value for the spherical Seidel coefficient. Then we use this value to synthetically generate a PSF and use it to deconvolve the blurry image. We then compute the sum of the spatial gradient of the resulting deconvolved image (this acts as a surrogate measure of image sharpness) and use its negative as a loss. We then minimize this loss (maximize the sharpness) by updating our initial guess of the spherical aberration coefficient using its gradient with respect to the loss function. Running this iteratively, we eventually converge to a final spherical aberration coefficient, generate a final synthetic PSF and deconvolve the blurry image with this PSF to get the final result.

Note that this procedure generalizes to spatially varying systems. We would instead jointly estimate all five Seidel coefficients and use ring deconvolution instead deconvolution at each step. This, however, is computationally expensive and requires generating $N$ (the image side length) PSFs per iterative step. We believe it is possible to do this more efficiently with DeepRD by replacing the ring deconvolution operation at each step with DeepRD. That is, we search the space of DeepRD networks for the one that produces the sharpest reconstruction; however, this is out of scope for this project and we leave it as future work.

## Experimental details
Experimental details for the micro-endoscopy experiment can be found in ref. 15.

**Sample preparation.** *Live tardigrades.* Tardigrades were mixed-staged adults (3–6 weeks old) of the eutardigrade species *Hypsibius exemplaris* Z151 (reclassified from *Hypsibius dujardini* in 2017), purchased from Sciento. Animals were cultured as described previously[63]. A mixture of starved and nonstarved tardigrades were stained overnight with Invitrogen nucleic acid gel fluorescent stain, whose excitation and emission maxima are 502 nm and 530 nm, respectively. Individual stained tardigrades were then isolated onto a glass slide for imaging. Meanwhile, the nonfluorescent samples (USAF resolution targets and rabbit liver tissue) were obtained imaging-ready on glass slides.

*BPAE cells.* BPAE cells were obtained from Thermo Fisher. They are labeled with MitoTracker Red CMXRos and Alexa Fluor 488 Phalloidin.

*Beads.* The 100-nm fluorescent beads were obtained from Polysciences (17150-10). For measuring the PSF in the light-sheet fluorescence microscope, we embedded the beads in 2% agarose with a final density $5 \times 10^{-4}$ of the stock solution.

*SU8686 cells.* SU8686 cells labeled with F-tractin-mRuby were embedded in soft bovine collagen and then fixed before imaging with light-sheet fluorescence microscope. They were obtained from ATCC.

**Imaging.** *UCLA Miniscope.* We used the UCLA Miniscope v.3 with the Ximea MU9PM-MBRD 12 bit, 2.2-µm pixel sensor. Optically, the Miniscope consists of a gradient-index objective and achromat tube lens; further details are provided elsewhere[1]. To obtain the system PSFs, we imaged 1-µm fluorescent beads randomly smeared on a glass slide. We used the resulting image to fit Seidel coefficients, obtaining 0.85, 0.56, 0.25, 0.29 and 0 waves of spherical aberration, coma, astigmatism, field curvature and distortion, respectively. These numbers, while specific to our particular assembly of the Miniscope, are consistent with the aberration profile of a radial gradient-index (GRIN) lens[64], which is the objective lens used by the Miniscope. The fact that the off-axis coefficients (all the primary coefficients except for spherical) are nonzero confirms that the system is indeed spatially varying. For comparison with standard deconvolution, we also acquired the center PSF by imaging a single fluorescent bead isolated and centered in the FoV. The PSF was then denoised before its use in deconvolution. For deconvolution microscopy calibration, we repeatedly diluted the bead solution with isopropyl alcohol until we were able to sufficiently isolate a single bead, whereas for RDM calibration we used a single dilution and imaged a slide containing a sparse collection of beads. We used a custom Prior Scientific 3D motion stage controlled with Micromanager v.1.4 and Pycromanager[65].

*Multicolor fluorescence microscope.* We used a Nikon Plan Apo VC ×100 Oil DIC N2 objective with 1.518 refractive index oil in a Nikon Eclipse Ti2 controlled with the Nikon NIS Elements Software (v.6.9.0). Images were taken with a Hamamatsu Orca Flash 4.0 camera with 0.065-µm pixel pitch. The PSFs were obtained with 0.01-µm FluoSpheres Yellow-Green505/515-nm F8803 and FluoSpheres Red 580/605-nm F8801 beads. First, we diluted beads in water and then further in ethanol until sufficient sparsity was achieved. The bead solution was then smeared on a slide and left to dry. Finally, the beads were mounted with a drop of glycerol and sealed with nail polish.

*Light-sheet fluorescence microscope.* We used a previously published setup for Field Synthesis[66] that was operated without a ring mask, rendering it to a multidirectional selective plane illumination mSPIM[67] system with a Gaussian sheet. The microscope equipped with 488 and 561 nm laser illumination, a Special Optics ×28.5/NA 0.67 illumination objective and a Nikon ×25/NA 1.1 detection objective, and is controlled with a custom LabView 2016 program written by Coleman Technologies and is equipped with temperature control for long-term live-cell imaging.

**Image processing.** All experimental images were captured and stored in a raw, unprocessed format (npy or tif). Miniscope images underwent hot pixel removal (detailed in the public code) and normalization before deblurring. These images were cropped afterward by

10 pixels in each dimension to remove edge artifacts. Multicolor images were downsampled by a factor of 2, separated into two channels and deblurred independently. After deblurring, the channels were recombined and globally contrast-stretched for display. Pseudocoloring was conducted with ImageJ using the Green/Magenta look-up table (LUT). These images were also cropped for edge artifacts. The details of the multimode fiber images can be found elsewhere[15]. The bead images were upsampled by 3× and convolved with a Gaussian kernel (3/2 pixel width) after deblurring. Neuron images were convolved with a Gaussian kernel (1 pixel width) after deblurring. This was carried out according to ref. 15. LSFM stacks were similarly cropped and contrast-stretched equally for each method for the purpose of display. For simulation data, images were normalized before deblurring and cropped after deblurring. All displayed PSFs were globally contrast-stretched for display.

**Computation.** PSF generation for the simulation experiments was conducted by synthetically generating pupil functions with the given Seidel coefficients[56]. Computation was conducted using Python on a single GPU, either a NVIDIA GeForce RTX 3090 or NVIDIA RTX A6000. For standard deconvolution the measured PSF was denoised through background subtraction and pixel-wise thresholding. For each 1,024 × 1,024 image from the Miniscope and high-NA multicolor systems, ring deconvolution took about 115 s and DeepRD took about 125 ms. For each 512 × 512 image in simulation ring deconvolution took about 60 s and DeepRD took about 0.1 s. For the 360 × 360 images from the micro-endoscope, ring deconvolution took about 20 seconds. For a single 512 × 512 × 160 volume from the LSFM system, sheet deconvolution took about 7 min.

All nonlearning, iterative methods are solving linear least squares optimization problems (equation (4)); we additionally add TV regularization to these and run them till convergence using an ADAM optimizer[58]. For each method, the hyperparameters (including learning rate and regularization strength) that provided the smallest loss and best qualitative results were used. For deconvolution we tried a variety of algorithms in addition to the iterative scheme, including Wiener filtering and Richardson–Lucy deconvolution, and used the best reconstruction, which was either iterative deconvolution or unsupervised Wiener filtering[68].

Open-source implementations of ring convolution, polar transform, Seidel fitting and ring deconvolution as well as the light-sheet extension methods can be found in our codebase. Our intent is for this codebase to function as an easy-to-use library such that any practitioner with any imaging system can utilize RDM with little-to-no overhead.

The baseline U-Net and DeepRD were both trained on ring-convolved images from the Div2k dataset. For the simulation results, both models were additionally fine-tuned on images from the CARE dataset. All models were trained till convergence of the validation loss and optimized over hyperparameters. The baseline U-Net architecture is based on the popular CARE model[21].

We used the following Python packages: Python 3.8.1, numpy 1.20.2, pytorch 2.4.1, scipy 1.6.2, scikit-image 0.17.2, pillow 8.2.0, matplotlib 3.2.2, tqdm 4.65.0, kornia 0.5.3 and jupyter 1.0.0. ImageJ 1.53a was also used for psuedocoloring for display.

**Reporting summary**
Further information on research design is available in the Nature Portfolio Reporting Summary linked to this article.

## Data availability
The data used in all of the imaging experiments (Miniscope, multicolor fluorescence, multimode fiber and light-sheet) are publicly available on Box (https://berkeley.box.com/s/zmsjjgmquwq2roh4d9qthcn-v3rhwuidn). Additional experimental data from the multimode fiber

system can be requested from ref.15 (https://opg.optica.org/boe/fulltext.cfm?uri=boe-11-8-4759&id=433935). The datasets used to train and fine-tune DeepRD and to evaluate the quantitative performance of the methods are also hosted on Box (https://berkeley.box.com/s/vv3g6avhrr9agijmlj3b1153oo7x9gao). These datasets were sourced from the CARE dataset[21] (https://publications.mpi-cbg.de/publications-sites/7207/) and the Div2k dataset[37] (https://data.vision.ee.ethz.ch/cvl/DIV2K/). The high-resolution pretraining dataset, due to its large memory useage, will be made available upon request.

## Code availability
The code for implementing ring convolution, ring deconvolution, DeepRD (including pretrained model weights) and Seidel PSF fitting along with tutorials on our experimental data are publicly available on GitHub (https://github.com/apsk14/rdmpy).

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

## Acknowledgements

We acknowledge A. Lyons and S. Kato's laboratory for providing the tardigrades, the authors of ref. 15 for providing the micro-endoscopy data, the Nikon Imaging Center at Harvard Medical School, N. Aggarwal for 3D printing, G. Gunjala, M. Gihana from the Danuser laboratory at the University of Texas Southwestern for SU8686 cells and D. Deb from the Janelia Research Campus for multi-GPU

experiments. A.K. was funded by the Berkeley Fellowship for Graduate Study and the Air Force Office of Scientific Research. A.N.A. was supported by the Berkeley Fellowship for Graduate Study and the National Science Foundation Graduate Research Fellowship Program under grant number DGE 1752814. Any opinions, findings, and conclusions or recommendations expressed in this material are those of the author(s) and do not necessarily reflect the views of the National Science Foundation. This work was funded by the Air Force Office of Scientific Research under award number FA9550-22-1-0521, CZI grant DAF2021-225666 and grant https://doi.org/10.37921/192752jrgbn from the Chan Zuckerberg Initiative DAF, an advised fund of Silicon Valley Community Foundation (funder https://doi.org/10.13039/100014989) and STROBE: A National Science Foundation Science and Technology Center under grant number DMR 1548924 (grant 1351896). L.W. is a Chan Zuckerberg Biohub SF investigator.

## Author contributions

A.K., A.N.A. and L.W. conceptualized the work. A.K. and A.N.A. developed the theory and code. A.K. created the light-sheet extension. A.K., D.M. and E.W. created the deep-learning component. A.K., A.N.A., S.Y., K.Y., F.M.G., B.C. and R.F. gathered experimental data. A.K., A.N.A., D.M., B.C., R.F. and L.W. wrote the paper.

## Competing interests

L.W. has a financial interest in SCI Microscopy. The other authors declare no competing interests.

## Additional information

**Extended data** is available for this paper at https://doi.org/10.1038/s41592-025-02684-5.

**Correspondence and requests for materials** should be addressed to Amit Kohli or Laura Waller.

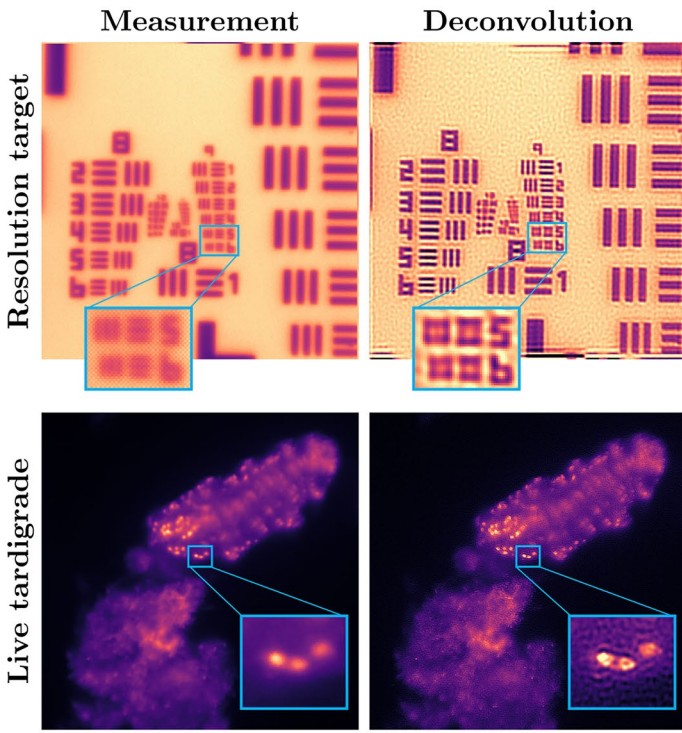

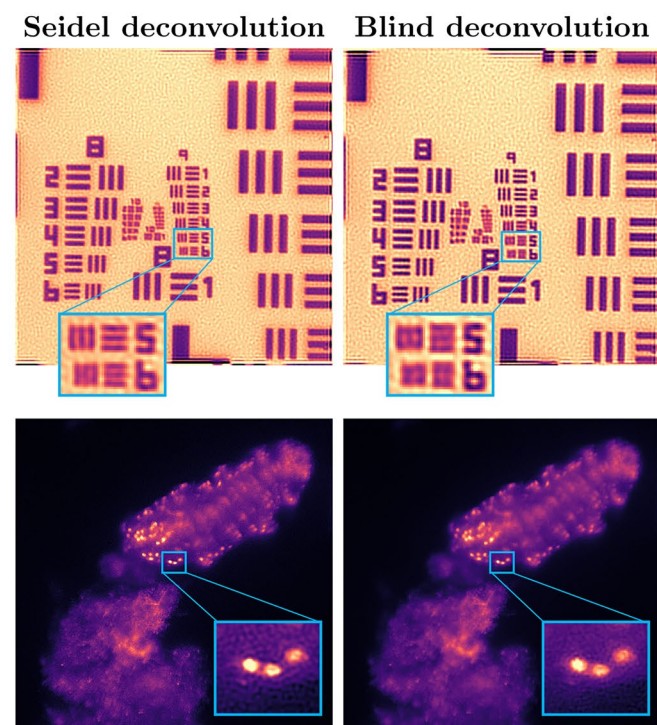

**Extended Data Fig. 1 | RDM for space-invariant systems.** Our Seidel and blind deconvolution algorithms compared with standard deconvolution on USAF test target and live tardigrade images from the Miniscope. The field-of-view is cropped to ensure space-invariance. Our methods outperform standard deconvolution by using a synthetic PSF, which prevents artifacts and loss of resolution from noise-based artifacts. Our blind deconvolution method usually correctly estimates the spherical Seidel coefficient well; however, for certain images, the blind method can overestimate the coefficient, leading to over sharpening of the image, as in the USAF resolution chart.

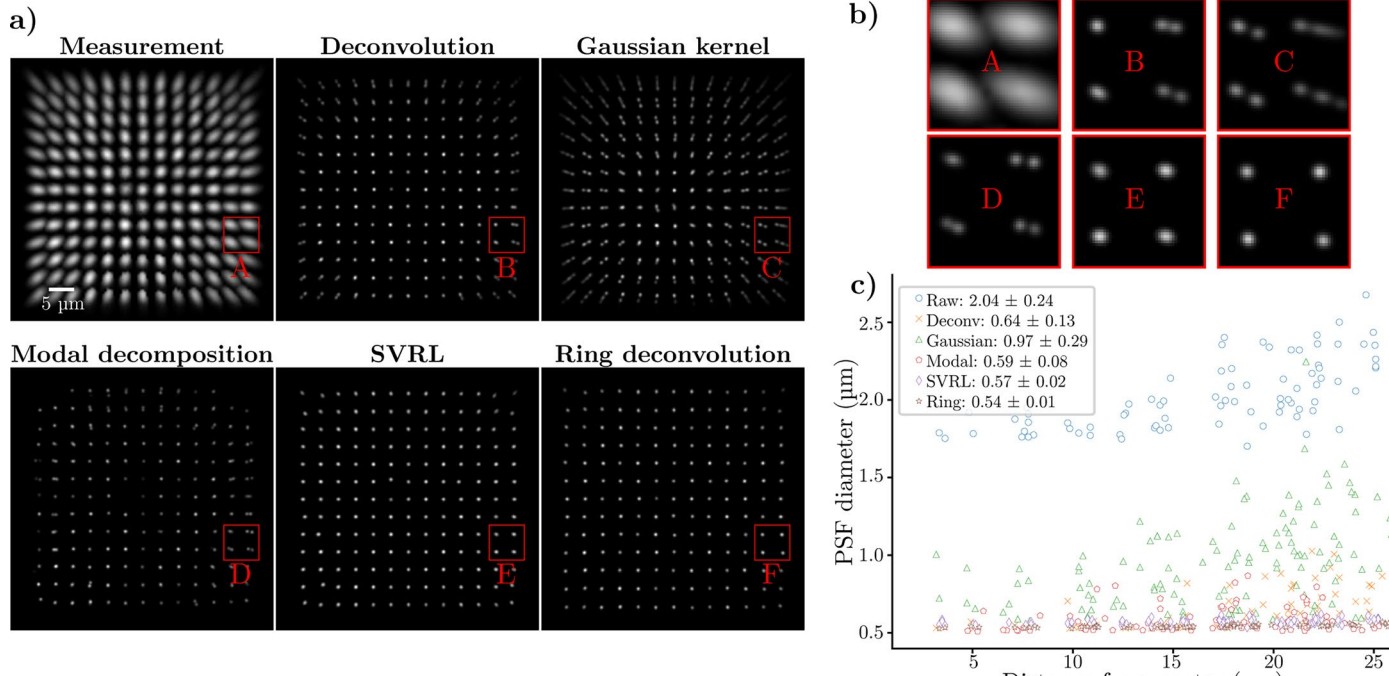

**Extended Data Fig. 2 | Additional comparisons on micro-endoscope data.**
An evenly spaced grid of 13 × 13 point sources are imaged with the micro-endoscope system. **a**) Results from each method. Top row (left to right) is the raw measurement, deconvolution with an experimental PSF and deconvolution with a Gaussian kernel fitted to the PSF. Bottom row (left to right) is deblurring via modal decomposition, SVRL, and ring deconvolution. As seen in **b**), the spatially varying methods (bottom row) are superior to the deconvolution methods (top row) due to the substantial spatial variation in the PSF. Among the spatially varying methods, ring deconvolution produces the smallest, most consistent beads. The quantitative results in **c**) also show that ring deconvolution has the smallest average bead radius with the least variation. Moreover, unlike the other methods, its performance does not degrade on beads far from the center of the FoV.

# Reporting Summary

## Statistics

For all statistical analyses, confirm that the following items are present in the figure legend, table legend, main text, or Methods section.

| n/a | Confirmed | |
|---|---|---|
| ☐ | ☒ | The exact sample size (*n*) for each experimental group/condition, given as a discrete number and unit of measurement |
| ☐ | ☒ | A statement on whether measurements were taken from distinct samples or whether the same sample was measured repeatedly |
| ☒ | ☐ | The statistical test(s) used AND whether they are one- or two-sided<br>*Only common tests should be described solely by name; describe more complex techniques in the Methods section.* |
| ☒ | ☐ | A description of all covariates tested |
| ☒ | ☐ | A description of any assumptions or corrections, such as tests of normality and adjustment for multiple comparisons |
| ☐ | ☒ | A full description of the statistical parameters including central tendency (e.g. means) or other basic estimates (e.g. regression coefficient) AND variation (e.g. standard deviation) or associated estimates of uncertainty (e.g. confidence intervals) |
| ☒ | ☐ | For null hypothesis testing, the test statistic (e.g. *F*, *t*, *r*) with confidence intervals, effect sizes, degrees of freedom and *P* value noted<br>*Give P values as exact values whenever suitable.* |
| ☒ | ☐ | For Bayesian analysis, information on the choice of priors and Markov chain Monte Carlo settings |
| ☒ | ☐ | For hierarchical and complex designs, identification of the appropriate level for tests and full reporting of outcomes |
| ☒ | ☐ | Estimates of effect sizes (e.g. Cohen's *d*, Pearson's *r*), indicating how they were calculated |

*Our web collection on statistics for biologists contains articles on many of the points above.*

## Software and code

Policy information about availability of computer code

| | |
|---|---|
| Data collection | Data was collecting using Micromanager v1.4 accompanied by Pycromanager, Nikon NIS Elements Software (version 6.9.0), and LabView 2016. |
| Data analysis | All code can be found in our github repository: https://github.com/apsk14/rdmpy<br>Python 3.8.1 was used with the following packages:<br>* numpy 1.20.2<br>* pytorch 2.4.1<br>* scipy 1.6.2<br>* scikit-image 0.17.2<br>* pillow 8.2.0<br>* matplotlib 3.2.2<br>* tqdm 4.65.0<br>* kornia 0.5.3<br>* jupyter 1.0.0<br><br>ImageJ  1.53a was also used for psuedocoloring for display |

For manuscripts utilizing custom algorithms or software that are central to the research but not yet described in published literature, software must be made available to editors and reviewers. We strongly encourage code deposition in a community repository (e.g. GitHub). See the Nature Portfolio guidelines for submitting code & software for further information.

## Data

Policy information about availability of data

All manuscripts must include a data availability statement. This statement should provide the following information, where applicable:

- Accession codes, unique identifiers, or web links for publicly available datasets
- A description of any restrictions on data availability
- For clinical datasets or third party data, please ensure that the statement adheres to our policy

The data used in all of the imaging experiments (Miniscope, multicolor fluorescence, multimode fiber, and light-sheet) is publicly available on Box (https://berkeley.box.com/s/zmsjjgmquwq2roh4d9qthcnv3rhwuidn). Additional experimental data from the multimode fiber system can be requested from Turcotte et al. (https://opg.optica.org/boe/fulltext.cfm?uri=boe-11-8-4759&id=433935). The datasets used to train and fine-tune DeepRD, and to evaluate the quantitative performance of the methods are also hosted on Box (https://berkeley.box.com/s/vv3g6avhrr9agijmlj3b1153oo7x9gao). These datasets were sourced from the CARE dataset Weigert et al. (https://publications.mpi-cbg.de/publications-sites/7207/) and the Div2k dataset Agustsson et al. (https://data.vision.ee.ethz.ch/cvl/DIV2K/). The high resolution pretraining dataset, due to its large memory useage, will be made available upon request.

## Human research participants

Policy information about studies involving human research participants and Sex and Gender in Research.

| | |
|---|---|
| Reporting on sex and gender | Not applicable |
| Population characteristics | Not applicable |
| Recruitment | Not applicable |
| Ethics oversight | Not applicable |

Note that full information on the approval of the study protocol must also be provided in the manuscript.

# Field-specific reporting

Please select the one below that is the best fit for your research. If you are not sure, read the appropriate sections before making your selection.

☒ Life sciences ☐ Behavioural & social sciences ☐ Ecological, evolutionary & environmental sciences

For a reference copy of the document with all sections, see nature.com/documents/nr-reporting-summary-flat.pdf

# Life sciences study design

All studies must disclose on these points even when the disclosure is negative.

| | |
|---|---|
| Sample size | All samples in this study were individually imaged and processed independently. The methods presented are not statistical in nature and no statistics over the samples were relevant or reported. Sample sizes were chosen manually by the authors to ensure a sufficient, non-redundant display of the method. |
| Data exclusions | No data was excluded from the manuscript |
| Replication | The data acquisition and processing protocol was done by different authors in different locations at different times. In each case, the method proved to be successful with consistent performance across the different acquisition environments and imaging modalities. Code is provided to reproduce our main results. |
| Randomization | Samples/organisms were selected in a psuedorandom fashion by iterating over a large batch of the sample/organism and selecting individual images that maximized biologically relevant content. Since the method did not report any statistical attributes of the data, there was no specific criteria for sample/organism selection. All model data was randomly split into training, validation, and test sets. |
| Blinding | Blinding was only relevant in the context of the DeepRD model, which was only tested on unseen experimental data or an unseen synthetic test set. |

# Reporting for specific materials, systems and methods

We require information from authors about some types of materials, experimental systems and methods used in many studies. Here, indicate whether each material, system or method listed is relevant to your study. If you are not sure if a list item applies to your research, read the appropriate section before selecting a response.

## Materials & experimental systems

| n/a | Involved in the study |
|-----|----------------------|
| ☒ ☐ | Antibodies |
| ☐ ☒ | Eukaryotic cell lines |
| ☒ ☐ | Palaeontology and archaeology |
| ☐ ☒ | Animals and other organisms |
| ☒ ☐ | Clinical data |
| ☒ ☐ | Dual use research of concern |

## Methods

| n/a | Involved in the study |
|-----|----------------------|
| ☒ ☐ | ChIP-seq |
| ☒ ☐ | Flow cytometry |
| ☒ ☐ | MRI-based neuroimaging |

## Eukaryotic cell lines

Policy information about cell lines and Sex and Gender in Research

| | |
|---|---|
| Cell line source(s) | The cells imaged were from the bovine pulmonary artery endothelial (BPAE) cell line (CVCL_4130) and were sourced from ThermoFisher (https://www.thermofisher.com/order/catalog/product/F36924). The other cells used were from the SU.86.86 cell line sourced from ATCC: CRL-1837 (https://www.atcc.org/products/crl-1837). |
| Authentication | Cells were authenticated by ThermoFisher. The authors did not authenticate them before imaging. |
| Mycoplasma contamination | To the author's knowledge, the cells were not tested for Mycoplasma contamination. |
| Commonly misidentified lines (See ICLAC register) | No commonly misidentified lines were used. |

## Animals and other research organisms

Policy information about studies involving animals; ARRIVE guidelines recommended for reporting animal research, and Sex and Gender in Research

| | |
|---|---|
| Laboratory animals | Mixed-staged adults of the eutardigrade species Hypsibius exemplaris Z151 (reclassified from Hypsibius dujardini in 2017), purchased from Sciento (Manchester, United Kingdom). Ages ranged from 3 to 6 weeks. |
| Wild animals | The study did not involve wild animals. |
| Reporting on sex | Sex-based analysis is not relevant for this study. The technique presented is about deblurring images taken with aberrated systems and is agnostic to the sample being imaged. |
| Field-collected samples | The study did not involve samples collected from the field. |
| Ethics oversight | No ethical approval or guidance is required for tardigrades. |

Note that full information on the approval of the study protocol must also be provided in the manuscript.

