## [Peer Review File · Nature Methods]

OVERVIEW

We would like to thank all three reviewers for their thoughtful comments and productive suggestions. We are glad the reviewers generally found the results to be of good quality and were appreciative of the transparency and reproducibility of our methods. The primary criticisms from reviewers have inspired major additions to the manuscript, which we feel have significantly improved the paper.

We first briefly summarize the main changes to the manuscript and how they address the central critiques. We then proceed to address each review in a point-by-point manner. For the point-by-point responses, the **blue text** indicates our response and the **black text** in bullet points are the reviewer's words verbatim.

Key Critiques:

1) **Better discussion of how RDM differs from related work**

R2 and R3 wanted to see a more clear discussion of how our method is different from previous literature that uses symmetry for spatially-varying deconvolution. We have added this and clarified some misconceptions about the previous work. Additionally, both R2 and R3 wanted to see an additional comparison with baselines; R2 specifically requested that we do so on a grid of PSFs from the micro-endoscope system and R3 asked for Gaussian kernel deconvolution and an additional spatially-varying method to compare to. We thank the reviewers for these suggestions and have implemented all of them.

2) **More evidence to support the generality of the method**

The reviewers felt that additional experiments were needed to aid the manuscript's generality and breadth. R1 specifically wanted to see how the method performed under Poisson noise. R2 and R3 both felt that RDM needs to be tested in the presence of higher-order aberrations. R2 and R3 also agreed that an extension beyond rotational symmetry would greatly strengthen the method's generality. In response, we have added new results addressing all of these: Light-sheet fluorescence microscopy (Sec. 2.4, 4.4, Fig. 6), higher-order aberration simulation (Appendix C.1, Fig. 10), simulation under Poisson noise (Appendix B, Fig. 9), and additional comparisons with existing methods (Appendix A, Fig. 8). The figures are shown on the next page and are detailed throughout the response.

Major changes:

1) **New experiments:**

Central to this resubmission is the addition of 4 new experiments.

- a) **Beyond rotational symmetry (Sec. 2.4, 4.4, Fig. 6).** First and foremost is the extension of RDM to non-rotationally symmetric systems. In particular, we adapt RDM to the laterally-symmetric case of light-sheet fluorescence microscopy

(LSFM). We derive a new forward model under this symmetry, a corresponding deblurring method, and an accompanying PSF model which can be fit from a single calibration volume. We rigorously test these algorithms on beads and cells imaged with a LSFM system and find that they significantly outperform 3D deconvolution.

b) **Higher-order aberrations (Appendix C.1, Fig. 10).** Next, we simulate RDM in the presence of higher-order aberrations and find that RDM using only the primary 5 Seidel coefficients still performs well. This aids the argument that RDM is robust and practical, and that the 5 primary Seidel coefficients are sufficient in practice.

c) **Poisson noise (Appendix B, Fig. 9).** We also simulate RDM under Poisson noise, finding little to no change in its performance compared to the case of deblurring under Gaussian noise.

d) **Baselines (Appendix A, Fig. 8).** Finally, we compare RDM, SVRL, and two new baselines—Gaussian kernel deconvolution and spatially-varying modal deconvolution—on a PSF grid from the micro-endoscope system. We find that RDM is both qualitatively and quantitatively the best among these methods.

REVIEWER 1

- The paper's clarity could be improved slightly and the simulated validation should be repeated with Poisson noise.
- Procedure is tested with simulated Gaussian noise. How does it work with Poisson noise?

We have made significant updates to the wording in the paper to try to make it more clear, and re-ran our simulated results with Poisson noise (see above and Appendix B), finding little difference in the results.

- Notation in the methods section is not adequately defined. For example, what does \star_{θ} in equation (2) mean? I'm guessing convolution over variable θ , but this is non-standard and is not defined.

You are correct, we added the following clarification directly after the equation. “The \star_{θ} operator indicates a one-dimensional convolution over the θ variable”.

- In introduction the ring convolution equation (2), it's worth mentioning integration by substitution. It's not immediately obvious where the leading radius r term came from.

Great suggestion, we added the following line prior to equation (2):

“By substituting (2) into (1) with $x = \rho \cos\phi$ and $y = \rho \sin\phi$, we get...”

And this line after eq (2):

“The r arises in the deconvolution since we are integrating over object space (u, v) in polar coordinates (r, θ) .”

- It's unclear how ground truth (GT) images of the beads were obtained for PSF estimation. If GT images aren't required (and the method instead assumes the beads were centered where they were observed in the distorted image) the proposed method seems unable to account for any tilt (translations) in the optical aberrations---such aberrations would be necessary to model barrel distortions and other common (symmetric) optical aberrations.

Ground truth images of the beads are never acquired, but the method does not assume the beads are centered either. The idea is to *estimate the center* from the observed measurement of randomly-scattered beads. In practice, we find the center is relatively easy to estimate heuristically using the vignetting effects and spatial variance of the system. Once the center is known, RDM can adjust to that center and thus correct for things like barrel distortion (though not tilt, since tilt is not a revolution-invariant aberration).

We add the following clarification in the Seidel fitting section (4.2):

“The locations of the point sources are not known a priori and are estimated via local peak detection. The optical center of the system is also needed in order to properly localize the PSFs; this can be found heuristically or by using common optical center finding algorithms such as [59].”

- pg 2: "While some existing deblurring..." is not a full sentence.

Thanks for finding this typo, we corrected it.

REVIEWER 2

- My main criticism is related to the RDM. The innovation of the method is not clear explained in the manuscript. I can't see the essential difference between RDM used in this manuscript and the radial deblurring used in the references [24-31]. It looks like they

used the same method, the only difference is the parameters in calculation. In this manuscript, the author used Seidel coefficients to represent PSFs. In other papers, they calculate PSF directly and also use ring deblurring to deconvolution. What are the advantages of introduce Seidel coefficients? I suggest the author give thoroughly and comprehensive description.

Thank you for this point, we can understand the confusion and have modified the manuscript to clearly explain the novelty of our work beyond refs [24-31]. Here is the new text added to the Introduction section:

“While some existing deblurring techniques have leveraged this symmetry, they are approximate and restricted to a specific subset of radially-varying blurs. [24, 25] only model blurs due to camera zoom. [26] uses log polar transforms to obtain an approximate space-invariant solution in the specific case of a parabolic mirror. [27–30] derive an approximate radially-varying scheme only for blurs from a single lens by applying deconvolution with theoretically-derived PSFs to 4 concentric regions assumed to be isoplanatic. [31] does the same for DSLR cameras and also requires 3 image channels from an RGB camera. In contrast, what we propose applies to *any* rotationally-symmetric imaging system, can incorporate more complex PSFs—even if they cannot be theoretically derived, makes no approximations (e.g., isoplanatic regions) in the image formation model, and can easily extend to other symmetries (such as in light-sheet microscopy).”

We also put this information in a table for a more direct view:

Paper	Works for any rotationally symmetric imaging system	Theoretically exact forward model	Microscopy experiments	Fourier analysis	Extends to other symmetries
[24]	✗ only zoom blur	✓ Only for zoom blur	✗ simulation only	✗	✗
[25]	✗ only zoom blur	✓ Only for zoom blur	✗ simulation only	✗	✗
[26]	✗ only parabolic mirrors	✗ Approximates as shift-invariant system	✗ microwave only	✗	✗
[27-30]	✗ only for a single spherical lens	✗ Does 2-4 radial patches	✗ Photography scale only	✗	✗
[31]	✗ Requires RGB channels	✗ Does 2-4 radial patches and is blind	✗ Photography scale only	✗	✗

RDM	✓ No additional assumptions	✓ Proof included	✓ 4 completely different microscope modalities with video and 3D.	✓ Fourier theorem provided along with filtering interpretation	✓ Use a similar PSF fitting strategy and derivation for light-sheet
-----	-----------------------------	------------------	---	--	---

Seidel coefficient fitting is more flexible than direct PSF calculation because it works for virtually any LRI imaging system. In contrast, direct PSF calculation can only be done for simple, idealized systems (e.g., the single lens or parabolic mirror from the other papers) that have analytical solutions and is sensitive to any mismatch between the idealized model and actual system. The systems we experiment with, for example, are too complicated for direct PSF calculation and require a calibration-based scheme like Seidel fitting to obtain accurate PSFs.

Our method introduces ring convolution and ring deconvolution; these methods are new in our work and do not appear elsewhere. We view these methods as a fundamental contribution to imaging because they are optimal under LRI; this is analogous to how standard deconvolution is optimal under LSI. The prior works have much more restrictive assumptions, or are heuristic/approximate. For example, [24,25] invert a different image formation model that only works for camera zoom with a linear velocity, [26] approximately suppresses the radial variance with logarithms and perform regular 2D deconvolution, and [27-31] computes a patchwise deconvolution with only a few concentric patches. None of these approaches are optimal for rotationally-symmetric imaging systems, while ring (de-)convolution is in theory.

- It would be beneficial if the applicability of RDM to confocal microscopy and non-linear imaging systems, such as two-photon and three-photon microscopy, could be explored. These systems, characterized by higher numerical apertures, present more significant spatial variant aberrations. Providing specific examples in these contexts would greatly enhance the manuscript.

We add a high numerical aperture light-sheet fluorescence microscopy experiment to the paper. Details for it are in Sec. 2.4 and come up shortly in this response document. We hope this will help demonstrate the broad applicability of our technique and its simplicity in transferring to other domains.

Our manuscript, post-revision, now contains experiments on miniature microscopy, high NA multicolor microscopy, micro-endoscopy, and light sheet-fluorescence microscopy. We feel that these experiments showcase the method's performance in a broad range of scenarios. Having said that, our codebase is generic and allows anyone to run our method on their own imaging system.

- In the section discussing multimode fiber endoscopy (Figure 4), the comparison with the SVRL method, which provided a PSF grid for the entire field of view post-deconvolution, was insightful. To strengthen your argument, a similar figure illustrating the performance improvements with RDM post-deconvolution would be valuable. Furthermore, the assertion that RDM surpasses SVRL in performance could be more compelling with a detailed comparison that includes optimal tuning of SVRL's hyper-parameters to achieve optimal performance and speed. And I think SVRL should be a benchmark in the following analysis. In addition, since SVRL's approach of decomposing the whole FOV's PSF allows for non-symmetric and non-uniform deblurring, it appears more versatile in handling spatial variant deconvolution. An expanded comparison, including aspects beyond speed and performance, would be informative.

This is a great suggestion; we did exactly as the reviewer suggests and added a new figure to the Supplement (see above and Appendix A) where we show SVRL, ring deconvolution, and few other additional baselines to a PSF grid. Using the PSF grid, we also quantify the improvement of ring deconvolution over SVRL finding that on average, it produces bead estimates with smaller full width half maxima and less standard deviation.

To clarify, SVRL's hyperparameters are already optimally tuned; the SVRL results we show were optimized and provided by Turcotte et al. We agree that an additional discussion comparing SVRL and ring deconvolution is beneficial and added it in the text.

- The potential application of RDM to systems beyond LRI, mentioned in section 2.4, lacks substantial supporting evidence. Presenting a compelling case study or data set could significantly enrich the manuscript's contributions.

We have put significant effort into extending our results to include linear lateral symmetry, rather than radial (see above, and Sections 2.4 and 4.4).

Regard as DeepRD, its model is based on Seidel coefficients from my understanding. From optical perspective, this is limited to low order aberration and missing high order ones. Therefore, in some extreme condition with larger aberration, it may be beyond this model's reach. This limitation becomes apparent in conditions with large aberrations, as shown in your results (Figure 5, g-h). And this may be the reason as you described DeepRD, "without guarantees on out of-distribution accuracy," in Page 3, paragraph 1. Why it is sufficed to characterize spatially-varying LRI systems? Exploration of adding higher orders would be informative?

- As described before, The suitability of DeepRD for systems with extremely large aberrations remains unclear, especially given its demonstrated efficacy in contexts with relatively low aberration and distortion, no matter in Multicolor fluorescence microscopy and Miniscope. Insights into DeepRD's performance in more challenging scenarios, such as multimode fiber endoscopy and high-NA microscopy, would be of great interest.

This is a good question. We conducted a new experiment in which we simulate a system that has higher-order aberrations (we used 6th order Seidel coefficients for a total of 13 coefficients) and perform the RDM procedure on it (see above, and Appendix C.1). First, we attempt to fit the 5 primary coefficients to the system's PSFs, which were generated with all 13 coefficients. The Seidel fitting algorithm converged to slightly inflated versions of the primary coefficients, compensating for the additional blur induced by the higher-orders. Then, using those fitted coefficients, DeepRD and ring deconvolution both achieve similar performance on the system than a system with no high-order coefficients. This reinforces our belief that RDM with just the 5 primary Seidel coefficients is robust to a large range of LRI blurs and thus sufficient for practical systems. Regardless, we do implement 6th order coefficients in our codebase, so users have the option of adding higher-order coefficients.

We would also like to directly clarify a few of the points brought up:

- While the 5 primary Seidel coefficients are limited to the dominant lower-order aberrations, there are theoretically infinite Seidel coefficients, which become increasingly higher order. As per the experiment above, we feel that the 5 primary coefficients are sufficient for practical application of RDM.
- We would also like to draw a distinction between large aberrations and high-order aberrations. "Large aberrations" means that the coefficient terms themselves are large, representing a significant presence of a particular aberration type. It is entirely possible, and in fact common, to have large aberrations that are primarily low-order. Similarly, it is also common to have high-order aberrations that are small. In Figure 5 we address large, lower-order aberrations (up to 3 waves for each coefficient). For context, the aberration profile in the multimodal fiber experiment ([0.9136 sphere, 0.0029 coma, 1.3004 astigmatism, 0.2229 field curvature, 1.2491 distortion] obtained from our Seidel fit) has at most 1.3 waves. Despite this challenging scenario, DeepRD performs well, outclassing deconvolution and the baseline U-Net. For example, DeepRD achieves a PSNR of 22.6 above its average (21.4) on an image that is blurred with extreme aberrations and distortion (coefficients [0.53 sphere, 0.77 coma, 0.91 astigmatism, 0.76 field curvature, 1.85 distortion]). While it does perform worse than ring deconvolution, this is expected since ring deconvolution is the gold standard analytical inverse method of the image formation model used to generate the dataset.
- Finally, our statement that DeepRD does not have guarantees for out of distribution data is a fact about deep learning models in general; despite any number of experiments, there is no way to guarantee that any deep learning model will perform well on an image from an arbitrary distribution. If an application requires absolute guarantees or is significantly out-of-distribution from the training data, one can use RDM instead of DeepRD. Or if speed/compute are not an issue, RDM should also be preferred. For example, the data from the

multimode fiber experiment has a small image size (120x120) and does not benefit from DeepRD since ring deconvolution runs within a few seconds.

- In Page 6, paragraph 2, this paper concludes that "In summary, we find that ring deconvolution is consistently the best method, a result that reflects the fact that it is theoretically exact for rotationally symmetric systems and is not biased toward any particular distribution of images.". Providing robust theoretical and experimental evidence to substantiate this claim would be highly beneficial.

We modify the statement to be more specific and nuanced:

"In summary, we find that ring deconvolution consistently produces the best reconstructions among the methods we tested on our diverse panel of imaging systems. We believe the improvement arises due to the lack of approximations and heuristics in the method's derivation. Moreover, as compared to deep-learning-based methods, ring deconvolution undergoes no learning procedure, and thus does not transmit bias from the training data into future reconstructions."

- The application of RDM and DeepRD to dynamically changing optical systems warrants further exploration. For example, the study's inclusion of multimode fiber imaging data, where the PSF dynamically shifts with the movement of the fiber probe in practical usage of such endoscopy (e.g. the two papers published on Nature Photonics 2023, 17, 679-687 and Nature Photonics 2015, 9, 529-535). Presenting results from such dynamic contexts would provide valuable insights into the versatility of these methods.

We agree with the reviewer that an exploration of dynamically changing optical systems with RDM and DeepRD would be a worthwhile future exploration; however, we feel that it is beyond the scope of this work. Instead, we have provided a variety of other new experiments to display RDM's versatility (see above).

- Figure 5 (i) indicates that SeidelNet has 6.9M parameters and Unet has 92M parameters. Despite SeidelNet having around 15 times fewer parameters, why does it have a longer runtime than Unet? I would like to confirm this once again.

This is correct, DeepRD has ~15x fewer parameters than the U-Net. The slower runtime of DeepRD is due to its architecture, which uses a Hypernetwork (a neural network whose output is another neural network). Hypernetworks have slower evaluation time than simpler architectures but allow for us to split DeepRD into a calibration step in which a system-specific deblurring network is obtained from Seidel coefficients, and a deblurring step where the deblurring network is used to deblur an image.

Fewer parameters are used in DeepRD to ensure that its performance improvement over the U-Net is not simply due to having more parameters. We tested multiple sizes of

U-Nets, including ones with the same and fewer parameters than DeepRD, and picked the best performing one.

We hope to keep updating the architecture of DeepRD over time as newer models are developed. We are in particular excited by the next generation of natural language processing models, which offer powerful new ways to condition on specific information and enable the model to focus on a particular subset of the data distribution.

REVIEWER 3

- Overall, this work demonstrates an interesting contribution to deconvolution microscopy, though the usage of PSF rotational symmetry has been explored before in other contexts [1-2]

This is a common misconception. In our work, the PSF is not rotationally symmetric, as in [1-2]. Instead, we are exploiting the rotational symmetry of the **optical system**. This yields PSFs that are **not rotationally symmetric**, and are space-varying. Instead, they satisfy a different property, called linear revolution invariance (LRI), that we define in this paper. LRI encompasses many common optical aberrations, such as coma, astigmatism, field curvature, and distortion, that do not result in rotationally-symmetric PSFs.

- The strong reliance on rotational symmetry is a significant limitation. While this assumption is valid for optical imaging systems at the PSF level, its application for deconvolving real images raises questions about its reliability. Consequently, RDM appears to be an advanced PSF approximation with constraints rather than a significant advancement in spatially varying deconvolution microscopy. The claim of providing an "Exact solution" may thus be overly ambitious without further clarification and justification.

To avoid any ambiguity, we would like to clarify that we **do not assume rotational symmetry at the PSF level**, but rather at the system level. See the previous comment. The reason we believe this is a significant advancement in spatially-varying deconvolution microscopy is that the method leverages commonly recognized symmetries in the spatially-varying PSF to more efficiently and accurately perform deblurring. In the presence of these symmetries, the method is indeed theoretically exact, in precisely the same sense that the convolutional imaging model is exact under linear shift invariance. Having said that, we have removed "exact solution" from the paper and title so as not to over-claim the method's practical performance.

In this new revision, our experiments demonstrate strong practical performance of the method across many imaging tasks, including miniature microscopy, high NA multicolor microscopy, micro-endoscopy, and light sheet-fluorescence microscopy. In our view, this

is more than enough evidence to claim that the method is a useful contribution to the literature on spatially-varying deconvolution.

In our new experiments, we have additionally extended our method to 3D symmetries beyond rotational symmetry in our light-sheet fluorescence microscopy experiment (see above, and Sections 2.4 and 4.4).

Furthermore, we would like to clarify that RDM is not an “advanced PSF approximation”. The main innovation in RDM is the derivation of the forward and inverse system model (ring convolution/deconvolution), which enable deblurring. **These are not PSF approximations. They are forward/inverse models.** Indeed, the method can run with PSFs that are acquired in any way, including experimentally or by theoretical modeling. We do utilize PSF approximation via Seidel fitting in the paper as a practical scheme to help with denoising and to make calibration easier, but it is not required for RDM.

- The manuscript outlines the problem of Ring convolution and includes derivations and Fourier analysis. However, these do not constitute rigorous proof that RDM ensures exact recovery. A more nuanced discussion on the limitations and applicability of RDM would be beneficial.

We agree that the language of ‘exact recovery’ by RDM is too strong, since ring convolution is non-invertible due to the diffraction-limit of the system. We have altered this language and replaced it with more nuanced terms. Our goal is to communicate the fact that ring convolution exactly matches the gold-standard full linear superposition integral under revolution-invariance, and ring deconvolution produces the optimal solution to inverting ring convolution (it is a convex optimization problem). We also add the following nuanced discussion on the limitations and applicability of RDM:

“As is the case with standard deconvolution, there are conditions for which ring convolution is not fully invertible, and consequently, ring deconvolution will not recover the sample exactly. The diffraction-limit, for example, manifests in PSFs whose frequency spectrum is bandlimited, rendering frequencies in the sample that are past the bandlimit irrecoverable. This can also happen if certain frequencies in the system's transfer function are below the noise floor. In such cases ring deconvolution, because it is convex, will return an estimate of the sample that is closest in l_2 norm to the true sample. Regularization can improve this result even further by leveraging prior knowledge about the sample in question.”

- The demonstrations focus predominantly on 2D image recovery. For individual patches the proposed method requires over 100 seconds, and also the authors indicate that full image recovery could take an impractical amount of time (hundreds of hours). While DeepRD may offer a solution, the real-world applicability of RDM, especially for video

and 3D data, appears limited. The effectiveness of RDM on data with non-symmetric aberrations along the z-axis in 3D data is also questionable.

RDM is primarily focused on 2D image recovery but can be easily extended to higher dimensions, namely video data as shown in the manuscript for the miniscope system and for 3D deconvolution, as shown in our new light-sheet fluorescence microscopy example. There has been a misunderstanding here, **ring deconvolution runs on full images (512x512) in about 60 seconds** and DeepRD does so in about a tenth of a second. The statement about 'impractical amount of time (hundreds of hours)' is if one attempts to solve the rigorous superposition integral without using the LRI assumption, **not for RDM**. We realize that this line may be confusing, and so we remove it from the paper. Instead, we clarify with following language:

“Ring deconvolution provides the best reconstruction, but it is also the largest and slowest of the methods tested, taking about 60 seconds for a full image. However, if we were to try to deblur the image rigorously without using RDM, it would take hundreds of hours, despite being theoretically equivalent to ring deconvolution. Thus ring deconvolution allows for relatively fast, accurate deblurring where it was once infeasible.”

Therefore, we feel that RDM has strong real-world applicability to 3D and video data as evidenced by our waterbear videos (about 1 hour reconstruction time for a full video) and LSFM reconstructions (about 7 minutes reconstruction time per volume). However, we expect these speeds to get dramatically faster over time since the RDM algorithms are extremely parallelizable. In fact, early experiments using multiple GPUs with the Jax framework has shown a 20-fold decrease in runtime for ring deconvolution. This means that ring deconvolution on a 1024x1024 image would take about a second.

Runtime of ring deconvolution in iterations per second as a function of the number of GPUs used. Experiments were run using Chromatix <https://github.com/chromatix-team/chromatix>.

For the general case of aberrations along z , RDM can simply be applied slice-by-slice, using new PSFs for each new z position. The exact calibration mechanism is dependent on the type of z -dependent aberrations, but if it were just defocus, then only a single Seidel coefficient would need to be updated per z -slice. However, if the 3D PSF is shift-invariant along the z -dimension, then it is possible to incorporate RDM for 3D deconvolution just like we did for sheet deconvolution in the new LSFM experiment. This would involve inverting a 3D version of ring convolution, which would amount to convolving cylindrical regions of the sample volume with cylindrical regions of the sample PSF. We leave this as a future direction.

- The experimental results provided do not sufficiently highlight RDM's advantages over existing methods. A more thorough comparison with various PSF approximations (e.g., simple gaussian kernel), deconvolution algorithms (especially spatially-varying ones), and under different noise conditions (to see how robust the proposed method is) is advisable. The selection of microscopy modalities for demonstration seems arbitrary, and their relevance to spatially varying aberration should be clarified.
- In addition, a comparison with conventional deconvolution algorithms using a standard Gaussian PSF kernel would be useful.

We agree that a more thorough comparison with existing methods would benefit the paper. Consequently, we include a new experiment which compares RDM both quantitatively and qualitatively against Gaussian kernel deconvolution, spatially-varying modal deconvolution, and spatially-varying Richardson Lucy (SVRL) from Turcotte et al. We run these comparisons on an experimental grid of beads from the micro-endoscopy multimode fiber system (see above, and Appendix A). This experiment allows us to use these other spatially-varying methods since a rigorous grid calibration was provided for this system (i.e., a PSF at every point in the FoV). It also serves as a quantitative test bed since we are able to measure the full width half maxima of the beads after reconstruction with each method. RDM outperforms all the other methods we test in this regard.

In addition to the deblurring performance, these other spatially-varying methods all require extensive and specific calibration (e.g., an equally-spaced grid of PSFs). In contrast, RDM can be run with a single, randomly generated calibration image.

The selection of microscope modalities was intended to cover a wide range of systems with different underlying mechanisms, but that all suffered spatially-varying aberrations. In the beginning of each subsection in Results we discuss why each of these systems is subject to spatial-variance. The Miniscope allowed us to show how RDM can improve systems that are constrained to simple, uncorrected optics by their application (i.e., miniature *in vivo* imaging). The high NA multicolor microscopy shows how RDM can push the FoV of more commonly-used, commercial systems and applies to multicolor

imaging. The micro-endoscope system shows how RDM can significantly improve even complex, non-traditional imaging as long as it preserves symmetry and demonstrates that RDM works equally well on point-scanning systems. Finally, the LSFM shows how RDM can extend to symmetries beyond rotational symmetry and can improve aberrations induced by illumination, not just by optics. We add the following statement to the paper:

“We demonstrate this on four diverse microscope modalities: miniature microscopy, multicolor fluorescence microscopy, multimode fiber micro-endoscopy, and light-sheet fluorescence microscopy. These modalities each contain different characteristics and imaging mechanisms that are representative of a large range of systems, thereby forming a comprehensive basis to demonstrate the wide applicability and practical relevance of RDM.”

- The manuscript is well-written, but its structure could be improved for better clarity, particularly in the sections detailing reconstruction algorithms and microscopy modalities. A more seamless transition between sections is suggested.

We appreciate the feedback and have introduced a few minor improvements throughout the manuscript for added clarity including the statement from the previous bullet point.

- The claim in Figure 6 that elements 5 and 6 are not resolvable by other methods needs verification, which I think not true.

We agree that the statement is too vague as written and have corrected it to:

“In the first row, ring deconvolution and DeepRD clearly resolve resolution target elements near the edges of the FoV, which are not as well resolved by the two other methods.”

- Consider revising the title better to reflect the focus on rotationally symmetric imaging systems.

We appreciate this suggestion and have changed the title of the paper to:

“Ring Deconvolution Microscopy: exploiting symmetry for efficient spatially-varying aberration correction”

- The results from real data are less convincing compared to those from simulated scenarios.

This is an expected trend for all computational methods and is largely due to model mismatch. In simulation, we assume systems are perfectly rotationally symmetric, but in

reality, misalignment, manufacturing errors, noise correlations, and other experimental sources of error lead to only approximate rotational symmetry. Consequently, we expect some degree of degradation from simulation to experiment, but still find that RDM produces useful experimental results that match the trends seen in simulation.

- An explanation of why the chosen five Seidel coefficients are adequate is necessary, especially in light of Figure 6, which shows persistent artifacts from Seidel deconvolution. Can additional coefficients offer improvements for these cases?

The artifacts in Figure 6 are due to noise, not Seidel deconvolution. The artifacts are even worse in the baseline deconvolution, which uses an experimental PSF. Having said that, it is true that only using five Seidel coefficients induces some approximation error.

We add a new experiment and section to the paper regarding higher-order coefficients (see above and Appendix C.1). In summary, we find that the primary coefficients perform well empirically in approximating higher-order aberrations. Our codebase allows users to easily change the number of Seidel coefficients used in the approximation for their specific use-case (we have found this does not significantly improve RDM's performance).

- Detailed information on the running time of the proposed algorithm under various conditions should be provided to assess its practicality.

All runtimes are subject to the computational resources at hand. We detail ours in the Computation subsection (4.6): computation was done using Python on a single GPU, either a NVIDIA GeForce RTX 3090 or NVIDIA RTX A6000. 512 x 512 images with ring deconvolution take about 60 seconds to deblur a full image. In the Miniscope and high NA microscopy experiments, ring deconvolution took ~115 seconds for a 1024x1024 image. In the multimode fiber experiment, ring deconvolution takes about ~3 seconds for a 120x120 image and ~20 seconds for a 360x360 image. Finally in the LSFM experiment, sheet deconvolution took about 7 minutes for a 512x512x160 volume. We add these details to Sec. 4.6.

.

OVERVIEW

We thank the editor and reviewers for their support of our manuscript and are grateful that the improvements made in the last review cycle have been recognized.

To address the remaining technical feedback, we have made the following changes to the manuscript:

- Added a new appendix, **Appendix B**, which quantifies the performance of our light-sheet calibration procedure and sheet deconvolution. This includes a new figure copied below, **Figure 9**

Figure 9: Quantifying Sheet deconvolution microscopy. a) Error evaluation of PSF fitting models. N=6 image stacks of randomly-scattered point sources are acquired with a light-sheet microscope and fit using either with the modified Gibson-Lanni model, by interpolating the measured PSFs, or by using the (denoised) center PSF and assuming shift-invariance. The PSNRs between the generated PSFs and the measured PSFs from the remaining calibration image stacks is computed for all three PSF models. These PSNRs are stratified by regions split along the light-sheet spread direction; center is where the light-sheet is the thinnest, while edge is where the light-sheet is the most spread. The resulting average PSNRs and their standard deviations in each region are plotted as a function of region. b) Forward model computation time for different data sizes. The runtimes of three different light-sheet forward models are plotted as a function of data size; true blur (manually superimposing the shift-varying PSFs at every point in the 3D volumes), sheet convolution, and standard 3D convolution. Times are acquired by processing randomly-generated 3D volumes and are averaged over 10 trials. c) Quantitative evaluation of sheet deconvolution. N=20 3D volumes of randomly sized and oriented ellipsoids are generated. These objects are blurred via true blur using PSFs generated from our modified Gibson-Lanni model, and are corrupted with Poisson noise (SNR 15) to simulate light-sheet measurements. The simulated measurements are deblurred with sheet deconvolution, iterative gradient-based deconvolution with total variation regularization, Richardson-Lucy deconvolution, and Wiener deconvolution. The process is repeated with increasing values of the β spread parameter (see 4.4.2), beginning with a system that is shift-invariant ranging to a system which is highly shift-variant due to rapid light-sheet spread. The average PSNR over all 20 objects for each method is plotted as a function of the spread parameter.

- Expanded the discussion to point towards future work in dynamic imaging.
- Made several minor organizational revisions and clarifications to language.

Below, we respond to the comments from each reviewer in turn.

Reviewer #1

R1: The proposed method is well-motivated, theoretically justified, and broadly applicable. I expect it will be widely used in microscopy and other imaging applications.

The revision is a significant improvement over the original submission. I especially appreciate the new light sheet results. All my concerns have been addressed and I support publication.

We appreciate the reviewer's support.

Reviewer #2

R2: The authors have addressed most of my concerns... I suggest accept the manuscript.

We appreciate the reviewer's support and respond to the remaining suggestions below.

R2: In this light-sheet fluorescence microscopy (LSFM) experiment, the authors applied Ring Deconvolution to address the issue of spatially varying aberrations in light-sheet microscopy. Since the thickness of the light sheet varies along the propagation axis (typically the z-axis within the field of view), causing resolution degradation at the edges, the authors effectively restored the resolution across the entire field of view, particularly in the edge regions, using ring deconvolution. The experimental details of the PSF calibration process could be appropriately supplemented, such as how to select point source locations, handle noise interference, and perform interpolation or fitting to generate the 3D PSF. Additionally, error evaluation should be included to verify the accuracy of the generated PSF.

Thanks for the suggestions, we address them as follows:

We have supplemented Section 4.4.2 to discuss additional information about the PSF calibration process. The new text is below:

In order to deploy the above models experimentally, one must ensure that the point sources are sufficiently sparse such that the PSFs are mostly non-overlapping. The exact point source locations are not important and can be estimated. Noisy PSFs are best handled with the Gibson-Lanni fitting method, which uses synthetic PSFs but still fits the measured PSFs well (see Appendix B). The interpolation method is more sensitive to noise but can be denoised effectively with thresholding and median filtering. The fitting procedures are detailed in our codebase, and follow the same general structure: first take a calibration stack of randomly-scattered point sources, estimate the locations of the PSFs using local peak finding algorithms, produce generated PSFs at those locations, and use the error between the generated and measured PSF to update the PSF model. In the case of interpolation, the last

step is replaced by linearly interpolating the two closest measured PSFs to the desired location to create the generated PSF there.

We verified the accuracy of the generated LFSM PSFs across the field-of-view. This is shown in the newly added Appendix B and Figure 9 (see main response).

R2: The analysis of Sheet Deconvolution's performance in light-sheet fluorescence microscopy is insufficiently detailed. While the experimental results highlight its resolution improvement and computational efficiency advantages, there is a lack of quantitative comparisons with standard 3D deconvolution methods. Metrics such as signal-to-noise ratio (SNR) and specific computation time or complexity analyses for different data sizes should be included to provide a more comprehensive evaluation.

We added a report on runtime over different data sizes and compared sheet deconvolution to standard 3D deconvolution techniques including iterative total variation deconvolution, Richardson-Lucy, and Wiener filtering. This is shown in the newly added Appendix B and Figure 9 (see above).

R2: I still suggest the authors discuss the validity of their methods under dynamic conditions (e.g., the two papers published on Nature Photonics 2023, 17, 679-687 and Nature Photonics 2015, 9, 529-535), and provide valuable insights into the versatility of these methods. They don't need to do experiments, but give some discussion.

We have added a discussion of the applicability of our methods in dynamic settings to the Discussion section of our manuscript. See the new text below:

RDM is well-suited to dynamic conditions in which the system or sample is changing, as long as the system calibration can be updated accordingly. For simpler cases that can be directly modeled, such as the time-varying deformation of an imaging fiber, it should be possible to update the initial calibrated PSFs with a theoretical model ([55,56]).

Reviewer #3

R3: The authors have made significant improvements compared to the previous version. The discussion on how RDM differs from related work and the additional results on various microscopy modalities are particularly helpful.

We thank the reviewer for their positive assessment. Our responses to the additional observations and comments are below.

R3: The advantages of the new concept of ring convolution are still not clearly presented. The visual differences between deconvolved images from RDM/Seidel/Sheet deconvolution and conventional methods seem minor.

The main benefit of RDM is improved image recovery in the presence of radially varying aberrations. Figure 5 gives a quantitative explanation of this improvement; see Figure 5 (e.g. deconvolution has a PSNR of 19, while RDM has a PSNR of 24).

On real data, although the PSNR/MSE cannot be calculated, our reconstructions do appear to be consistently better than deconvolution; the insets we have placed on Figures 1-7 make these improvements visible to the naked eye. The degree of improvement is up to the reader, but, we believe Figure 2 clearly shows a noticeably better reconstruction of an airforce target, rabbit liver, and live tardigrade.

R3: The necessity of DeepRD needs further justification because:

1. its performance appears to be significantly worse than that of ring deconvolution; and
2. the claimed faster processing time, which is a major motivation, is not demonstrated. It would be more convincing if this more advanced version offered additional advantages (I understand the interpretability of DeepRD is briefly discussed).

The faster processing time of DeepRD is demonstrated in Figure 5(i). (It is possible this was missed due to a typo on our end.) We have fixed the typo and included the adjusted figure below.

Here, it is shown that DeepRD is almost three orders of magnitude faster than RDM.

We have also added the following sentence in our discussion of speed:

DeepRD is almost three orders of magnitude faster than RDM.

R3: As an image restoration method for deconvolution microscopy, I still believe it is essential to comprehensively compare this approach with other state-of-the-art methods, such as CARE (15, 1090–1097 (2018)), even if the forward model might be slightly different. This will also assist in justifying the new concept of RDM.

We agree. The U-Net baseline in Figure 2 is the CARE baseline suggested by the reviewer. It is the CARE model trained on our data. The performance is worse than RDM.

R3: I cannot agree with the assertion that we should accept that the method generalizes “surprisingly” well for real data when trained on simulated data, as stated on page 6.

We removed this claim.

R3: The results obtained from the demo code on real data (<https://github.com/apsk14/rdmpy/blob/main/demo.ipynb>), last example, live tardigrade?) appear to exhibit serious artifacts that do not represent true tissues after ring deconvolution. These results also differ from those presented in the paper. Maybe there are some parameters to be tuned?

The code is very well-structured, making it easy to follow and understand. However, as mentioned, the results obtained from the demo code on real data (demo.ipynb) appear to exhibit serious artifacts that do not represent true tissues after ring deconvolution and DeepRD. These results also differ from those presented in the paper.

Thank you for going through the code in detail! The section you reference in the demo notebook is actually using a different sample from the one shown in the paper. It is worth noting also that the sample shown in the demo notebook has significant amounts of algae in it (in addition to the tardigrade), which may look like artifacts but is in fact part of the sample. For additional clarity, we added new cells to the notebook which run ring deconvolution and DeepRD on the same sample that we use in the paper; as expected, the output matches exactly the images presented in the paper.

R3: On page 18, the phrase “with a final density 5×10^{-4} of the stock” should be corrected to “ 5×10^{-4} ”.

Thank you; we fixed the typo.

R3: The distinction between sheet deconvolution and ring deconvolution is noted, but it is unclear why this should be included in a paper focused on ring deconvolution microscopy, even though the theory of rotational symmetry may be relevant.

Thank you for the thoughtful point. We view sheet deconvolution as a natural and practical extension of ring deconvolution.

OVERVIEW

We would like to thank all three reviewers for their thoughtful comments and productive suggestions. We are glad the reviewers generally found the results to be of good quality and were appreciative of the transparency and reproducibility of our methods. The primary criticisms from reviewers have inspired major additions to the manuscript, which we feel have significantly improved the paper.

We first briefly summarize the main changes to the manuscript and how they address the central critiques. We then proceed to address each review in a point-by-point manner. For the point-by-point responses, the **blue text** indicates our response and the **black text** in bullet points are the reviewer's words verbatim.

Key Critiques:

1) **Better discussion of how RDM differs from related work**

R2 and R3 wanted to see a more clear discussion of how our method is different from previous literature that uses symmetry for spatially-varying deconvolution. We have added this and clarified some misconceptions about the previous work. Additionally, both R2 and R3 wanted to see an additional comparison with baselines; R2 specifically requested that we do so on a grid of PSFs from the micro-endoscope system and R3 asked for Gaussian kernel deconvolution and an additional spatially-varying method to compare to. We thank the reviewers for these suggestions and have implemented all of them.

2) **More evidence to support the generality of the method**

The reviewers felt that additional experiments were needed to aid the manuscript's generality and breadth. R1 specifically wanted to see how the method performed under Poisson noise. R2 and R3 both felt that RDM needs to be tested in the presence of higher-order aberrations. R2 and R3 also agreed that an extension beyond rotational symmetry would greatly strengthen the method's generality. In response, we have added new results addressing all of these: Light-sheet fluorescence microscopy (Sec. 2.4, 4.4, Fig. 6), higher-order aberration simulation (Appendix C.1, Fig. 10), simulation under Poisson noise (Appendix B, Fig. 9), and additional comparisons with existing methods (Appendix A, Fig. 8). The figures are shown on the next page and are detailed throughout the response.

Major changes:

1) **New experiments:**

Central to this resubmission is the addition of 4 new experiments.

- a) **Beyond rotational symmetry (Sec. 2.4, 4.4, Fig. 6).** First and foremost is the extension of RDM to non-rotationally symmetric systems. In particular, we adapt RDM to the laterally-symmetric case of light-sheet fluorescence microscopy

(LSFM). We derive a new forward model under this symmetry, a corresponding deblurring method, and an accompanying PSF model which can be fit from a single calibration volume. We rigorously test these algorithms on beads and cells imaged with a LSFM system and find that they significantly outperform 3D deconvolution.

b) **Higher-order aberrations (Appendix C.1, Fig. 10).** Next, we simulate RDM in the presence of higher-order aberrations and find that RDM using only the primary 5 Seidel coefficients still performs well. This aids the argument that RDM is robust and practical, and that the 5 primary Seidel coefficients are sufficient in practice.

c) **Poisson noise (Appendix B, Fig. 9).** We also simulate RDM under Poisson noise, finding little to no change in its performance compared to the case of deblurring under Gaussian noise.

d) **Baselines (Appendix A, Fig. 8).** Finally, we compare RDM, SVRL, and two new baselines—Gaussian kernel deconvolution and spatially-varying modal deconvolution—on a PSF grid from the micro-endoscope system. We find that RDM is both qualitatively and quantitatively the best among these methods.

REVIEWER 1

- The paper's clarity could be improved slightly and the simulated validation should be repeated with Poisson noise.
- Procedure is tested with simulated Gaussian noise. How does it work with Poisson noise?

We have made significant updates to the wording in the paper to try to make it more clear, and re-ran our simulated results with Poisson noise (see above and Appendix B), finding little difference in the results.

- Notation in the methods section is not adequately defined. For example, what does \star_{θ} in equation (2) mean? I'm guessing convolution over variable θ , but this is non-standard and is not defined.

You are correct, we added the following clarification directly after the equation. “The \star_{θ} operator indicates a one-dimensional convolution over the θ variable”.

- In introduction the ring convolution equation (2), it's worth mentioning integration by substitution. It's not immediately obvious where the leading radius r term came from.

Great suggestion, we added the following line prior to equation (2):

“By substituting (2) into (1) with $x = \rho \cos\phi$ and $y = \rho \sin\phi$, we get...”

And this line after eq (2):

“The r arises in the deconvolution since we are integrating over object space (u, v) in polar coordinates (r, θ) .”

- It's unclear how ground truth (GT) images of the beads were obtained for PSF estimation. If GT images aren't required (and the method instead assumes the beads were centered where they were observed in the distorted image) the proposed method seems unable to account for any tilt (translations) in the optical aberrations---such aberrations would be necessary to model barrel distortions and other common (symmetric) optical aberrations.

Ground truth images of the beads are never acquired, but the method does not assume the beads are centered either. The idea is to *estimate the center* from the observed measurement of randomly-scattered beads. In practice, we find the center is relatively easy to estimate heuristically using the vignetting effects and spatial variance of the system. Once the center is known, RDM can adjust to that center and thus correct for things like barrel distortion (though not tilt, since tilt is not a revolution-invariant aberration).

We add the following clarification in the Seidel fitting section (4.2):

“The locations of the point sources are not known a priori and are estimated via local peak detection. The optical center of the system is also needed in order to properly localize the PSFs; this can be found heuristically or by using common optical center finding algorithms such as [59].”

- pg 2: "While some existing deblurring..." is not a full sentence.

Thanks for finding this typo, we corrected it.

REVIEWER 2

- My main criticism is related to the RDM. The innovation of the method is not clear explained in the manuscript. I can't see the essential difference between RDM used in this manuscript and the radial deblurring used in the references [24-31]. It looks like they

used the same method, the only difference is the parameters in calculation. In this manuscript, the author used Seidel coefficients to represent PSFs. In other papers, they calculate PSF directly and also use ring deblurring to deconvolution. What are the advantages of introduce Seidel coefficients? I suggest the author give thoroughly and comprehensive description.

Thank you for this point, we can understand the confusion and have modified the manuscript to clearly explain the novelty of our work beyond refs [24-31]. Here is the new text added to the Introduction section:

“While some existing deblurring techniques have leveraged this symmetry, they are approximate and restricted to a specific subset of radially-varying blurs. [24, 25] only model blurs due to camera zoom. [26] uses log polar transforms to obtain an approximate space-invariant solution in the specific case of a parabolic mirror. [27–30] derive an approximate radially-varying scheme only for blurs from a single lens by applying deconvolution with theoretically-derived PSFs to 4 concentric regions assumed to be isoplanatic. [31] does the same for DSLR cameras and also requires 3 image channels from an RGB camera. In contrast, what we propose applies to *any* rotationally-symmetric imaging system, can incorporate more complex PSFs—even if they cannot be theoretically derived, makes no approximations (e.g., isoplanatic regions) in the image formation model, and can easily extend to other symmetries (such as in light-sheet microscopy).”

We also put this information in a table for a more direct view:

Paper	Works for any rotationally symmetric imaging system	Theoretically exact forward model	Microscopy experiments	Fourier analysis	Extends to other symmetries
[24]	✗ only zoom blur	✓ Only for zoom blur	✗ simulation only	✗	✗
[25]	✗ only zoom blur	✓ Only for zoom blur	✗ simulation only	✗	✗
[26]	✗ only parabolic mirrors	✗ Approximates as shift-invariant system	✗ microwave only	✗	✗
[27-30]	✗ only for a single spherical lens	✗ Does 2-4 radial patches	✗ Photography scale only	✗	✗
[31]	✗ Requires RGB channels	✗ Does 2-4 radial patches and is blind	✗ Photography scale only	✗	✗

RDM	✓ No additional assumptions	✓ Proof included	✓ 4 completely different microscope modalities with video and 3D.	✓ Fourier theorem provided along with filtering interpretation	✓ Use a similar PSF fitting strategy and derivation for light-sheet
-----	-----------------------------	------------------	---	--	---

Seidel coefficient fitting is more flexible than direct PSF calculation because it works for virtually any LRI imaging system. In contrast, direct PSF calculation can only be done for simple, idealized systems (e.g., the single lens or parabolic mirror from the other papers) that have analytical solutions and is sensitive to any mismatch between the idealized model and actual system. The systems we experiment with, for example, are too complicated for direct PSF calculation and require a calibration-based scheme like Seidel fitting to obtain accurate PSFs.

Our method introduces ring convolution and ring deconvolution; these methods are new in our work and do not appear elsewhere. We view these methods as a fundamental contribution to imaging because they are optimal under LRI; this is analogous to how standard deconvolution is optimal under LSI. The prior works have much more restrictive assumptions, or are heuristic/approximate. For example, [24,25] invert a different image formation model that only works for camera zoom with a linear velocity, [26] approximately suppresses the radial variance with logarithms and perform regular 2D deconvolution, and [27-31] computes a patchwise deconvolution with only a few concentric patches. None of these approaches are optimal for rotationally-symmetric imaging systems, while ring (de-)convolution is in theory.

- It would be beneficial if the applicability of RDM to confocal microscopy and non-linear imaging systems, such as two-photon and three-photon microscopy, could be explored. These systems, characterized by higher numerical apertures, present more significant spatial variant aberrations. Providing specific examples in these contexts would greatly enhance the manuscript.

We add a high numerical aperture light-sheet fluorescence microscopy experiment to the paper. Details for it are in Sec. 2.4 and come up shortly in this response document. We hope this will help demonstrate the broad applicability of our technique and its simplicity in transferring to other domains.

Our manuscript, post-revision, now contains experiments on miniature microscopy, high NA multicolor microscopy, micro-endoscopy, and light sheet-fluorescence microscopy. We feel that these experiments showcase the method's performance in a broad range of scenarios. Having said that, our codebase is generic and allows anyone to run our method on their own imaging system.

- In the section discussing multimode fiber endoscopy (Figure 4), the comparison with the SVRL method, which provided a PSF grid for the entire field of view post-deconvolution, was insightful. To strengthen your argument, a similar figure illustrating the performance improvements with RDM post-deconvolution would be valuable. Furthermore, the assertion that RDM surpasses SVRL in performance could be more compelling with a detailed comparison that includes optimal tuning of SVRL's hyper-parameters to achieve optimal performance and speed. And I think SVRL should be a benchmark in the following analysis. In addition, since SVRL's approach of decomposing the whole FOV's PSF allows for non-symmetric and non-uniform deblurring, it appears more versatile in handling spatial variant deconvolution. An expanded comparison, including aspects beyond speed and performance, would be informative.

This is a great suggestion; we did exactly as the reviewer suggests and added a new figure to the Supplement (see above and Appendix A) where we show SVRL, ring deconvolution, and few other additional baselines to a PSF grid. Using the PSF grid, we also quantify the improvement of ring deconvolution over SVRL finding that on average, it produces bead estimates with smaller full width half maxima and less standard deviation.

To clarify, SVRL's hyperparameters are already optimally tuned; the SVRL results we show were optimized and provided by Turcotte et al. We agree that an additional discussion comparing SVRL and ring deconvolution is beneficial and added it in the text.

- The potential application of RDM to systems beyond LRI, mentioned in section 2.4, lacks substantial supporting evidence. Presenting a compelling case study or data set could significantly enrich the manuscript's contributions.

We have put significant effort into extending our results to include linear lateral symmetry, rather than radial (see above, and Sections 2.4 and 4.4).

Regard as DeepRD, its model is based on Seidel coefficients from my understanding. From optical perspective, this is limited to low order aberration and missing high order ones. Therefore, in some extreme condition with larger aberration, it may be beyond this model's reach. This limitation becomes apparent in conditions with large aberrations, as shown in your results (Figure 5, g-h). And this may be the reason as you described DeepRD, "without guarantees on out of-distribution accuracy," in Page 3, paragraph 1. Why it is sufficed to characterize spatially-varying LRI systems? Exploration of adding higher orders would be informative?

- As described before, The suitability of DeepRD for systems with extremely large aberrations remains unclear, especially given its demonstrated efficacy in contexts with relatively low aberration and distortion, no matter in Multicolor fluorescence microscopy and Miniscope. Insights into DeepRD's performance in more challenging scenarios, such as multimode fiber endoscopy and high-NA microscopy, would be of great interest.

This is a good question. We conducted a new experiment in which we simulate a system that has higher-order aberrations (we used 6th order Seidel coefficients for a total of 13 coefficients) and perform the RDM procedure on it (see above, and Appendix C.1). First, we attempt to fit the 5 primary coefficients to the system's PSFs, which were generated with all 13 coefficients. The Seidel fitting algorithm converged to slightly inflated versions of the primary coefficients, compensating for the additional blur induced by the higher-orders. Then, using those fitted coefficients, DeepRD and ring deconvolution both achieve similar performance on the system than a system with no high-order coefficients. This reinforces our belief that RDM with just the 5 primary Seidel coefficients is robust to a large range of LRI blurs and thus sufficient for practical systems. Regardless, we do implement 6th order coefficients in our codebase, so users have the option of adding higher-order coefficients.

We would also like to directly clarify a few of the points brought up:

- While the 5 primary Seidel coefficients are limited to the dominant lower-order aberrations, there are theoretically infinite Seidel coefficients, which become increasingly higher order. As per the experiment above, we feel that the 5 primary coefficients are sufficient for practical application of RDM.
- We would also like to draw a distinction between large aberrations and high-order aberrations. "Large aberrations" means that the coefficient terms themselves are large, representing a significant presence of a particular aberration type. It is entirely possible, and in fact common, to have large aberrations that are primarily low-order. Similarly, it is also common to have high-order aberrations that are small. In Figure 5 we address large, lower-order aberrations (up to 3 waves for each coefficient). For context, the aberration profile in the multimodal fiber experiment ([0.9136 sphere, 0.0029 coma, 1.3004 astigmatism, 0.2229 field curvature, 1.2491 distortion] obtained from our Seidel fit) has at most 1.3 waves. Despite this challenging scenario, DeepRD performs well, outclassing deconvolution and the baseline U-Net. For example, DeepRD achieves a PSNR of 22.6 above its average (21.4) on an image that is blurred with extreme aberrations and distortion (coefficients [0.53 sphere, 0.77 coma, 0.91 astigmatism, 0.76 field curvature, 1.85 distortion]). While it does perform worse than ring deconvolution, this is expected since ring deconvolution is the gold standard analytical inverse method of the image formation model used to generate the dataset.
- Finally, our statement that DeepRD does not have guarantees for out of distribution data is a fact about deep learning models in general; despite any number of experiments, there is no way to guarantee that any deep learning model will perform well on an image from an arbitrary distribution. If an application requires absolute guarantees or is significantly out-of-distribution from the training data, one can use RDM instead of DeepRD. Or if speed/compute are not an issue, RDM should also be preferred. For example, the data from the

multimode fiber experiment has a small image size (120x120) and does not benefit from DeepRD since ring deconvolution runs within a few seconds.

- In Page 6, paragraph 2, this paper concludes that "In summary, we find that ring deconvolution is consistently the best method, a result that reflects the fact that it is theoretically exact for rotationally symmetric systems and is not biased toward any particular distribution of images.". Providing robust theoretical and experimental evidence to substantiate this claim would be highly beneficial.

We modify the statement to be more specific and nuanced:

"In summary, we find that ring deconvolution consistently produces the best reconstructions among the methods we tested on our diverse panel of imaging systems. We believe the improvement arises due to the lack of approximations and heuristics in the method's derivation. Moreover, as compared to deep-learning-based methods, ring deconvolution undergoes no learning procedure, and thus does not transmit bias from the training data into future reconstructions."

- The application of RDM and DeepRD to dynamically changing optical systems warrants further exploration. For example, the study's inclusion of multimode fiber imaging data, where the PSF dynamically shifts with the movement of the fiber probe in practical usage of such endoscopy (e.g. the two papers published on Nature Photonics 2023, 17, 679-687 and Nature Photonics 2015, 9, 529-535). Presenting results from such dynamic contexts would provide valuable insights into the versatility of these methods.

We agree with the reviewer that an exploration of dynamically changing optical systems with RDM and DeepRD would be a worthwhile future exploration; however, we feel that it is beyond the scope of this work. Instead, we have provided a variety of other new experiments to display RDM's versatility (see above).

- Figure 5 (i) indicates that SeidelNet has 6.9M parameters and Unet has 92M parameters. Despite SeidelNet having around 15 times fewer parameters, why does it have a longer runtime than Unet? I would like to confirm this once again.

This is correct, DeepRD has ~15x fewer parameters than the U-Net. The slower runtime of DeepRD is due to its architecture, which uses a Hypernetwork (a neural network whose output is another neural network). Hypernetworks have slower evaluation time than simpler architectures but allow for us to split DeepRD into a calibration step in which a system-specific deblurring network is obtained from Seidel coefficients, and a deblurring step where the deblurring network is used to deblur an image.

Fewer parameters are used in DeepRD to ensure that its performance improvement over the U-Net is not simply due to having more parameters. We tested multiple sizes of

U-Nets, including ones with the same and fewer parameters than DeepRD, and picked the best performing one.

We hope to keep updating the architecture of DeepRD over time as newer models are developed. We are in particular excited by the next generation of natural language processing models, which offer powerful new ways to condition on specific information and enable the model to focus on a particular subset of the data distribution.

REVIEWER 3

- Overall, this work demonstrates an interesting contribution to deconvolution microscopy, though the usage of PSF rotational symmetry has been explored before in other contexts [1-2]

This is a common misconception. In our work, the PSF is not rotationally symmetric, as in [1-2]. Instead, we are exploiting the rotational symmetry of the **optical system**. This yields PSFs that are **not rotationally symmetric**, and are space-varying. Instead, they satisfy a different property, called linear revolution invariance (LRI), that we define in this paper. LRI encompasses many common optical aberrations, such as coma, astigmatism, field curvature, and distortion, that do not result in rotationally-symmetric PSFs.

- The strong reliance on rotational symmetry is a significant limitation. While this assumption is valid for optical imaging systems at the PSF level, its application for deconvolving real images raises questions about its reliability. Consequently, RDM appears to be an advanced PSF approximation with constraints rather than a significant advancement in spatially varying deconvolution microscopy. The claim of providing an "Exact solution" may thus be overly ambitious without further clarification and justification.

To avoid any ambiguity, we would like to clarify that we **do not assume rotational symmetry at the PSF level**, but rather at the system level. See the previous comment. The reason we believe this is a significant advancement in spatially-varying deconvolution microscopy is that the method leverages commonly recognized symmetries in the spatially-varying PSF to more efficiently and accurately perform deblurring. In the presence of these symmetries, the method is indeed theoretically exact, in precisely the same sense that the convolutional imaging model is exact under linear shift invariance. Having said that, we have removed "exact solution" from the paper and title so as not to over-claim the method's practical performance.

In this new revision, our experiments demonstrate strong practical performance of the method across many imaging tasks, including miniature microscopy, high NA multicolor microscopy, micro-endoscopy, and light sheet-fluorescence microscopy. In our view, this

is more than enough evidence to claim that the method is a useful contribution to the literature on spatially-varying deconvolution.

In our new experiments, we have additionally extended our method to 3D symmetries beyond rotational symmetry in our light-sheet fluorescence microscopy experiment (see above, and Sections 2.4 and 4.4).

Furthermore, we would like to clarify that RDM is not an “advanced PSF approximation”. The main innovation in RDM is the derivation of the forward and inverse system model (ring convolution/deconvolution), which enable deblurring. **These are not PSF approximations. They are forward/inverse models.** Indeed, the method can run with PSFs that are acquired in any way, including experimentally or by theoretical modeling. We do utilize PSF approximation via Seidel fitting in the paper as a practical scheme to help with denoising and to make calibration easier, but it is not required for RDM.

- The manuscript outlines the problem of Ring convolution and includes derivations and Fourier analysis. However, these do not constitute rigorous proof that RDM ensures exact recovery. A more nuanced discussion on the limitations and applicability of RDM would be beneficial.

We agree that the language of ‘exact recovery’ by RDM is too strong, since ring convolution is non-invertible due to the diffraction-limit of the system. We have altered this language and replaced it with more nuanced terms. Our goal is to communicate the fact that ring convolution exactly matches the gold-standard full linear superposition integral under revolution-invariance, and ring deconvolution produces the optimal solution to inverting ring convolution (it is a convex optimization problem). We also add the following nuanced discussion on the limitations and applicability of RDM:

“As is the case with standard deconvolution, there are conditions for which ring convolution is not fully invertible, and consequently, ring deconvolution will not recover the sample exactly. The diffraction-limit, for example, manifests in PSFs whose frequency spectrum is bandlimited, rendering frequencies in the sample that are past the bandlimit irrecoverable. This can also happen if certain frequencies in the system's transfer function are below the noise floor. In such cases ring deconvolution, because it is convex, will return an estimate of the sample that is closest in l_2 norm to the true sample. Regularization can improve this result even further by leveraging prior knowledge about the sample in question.”

- The demonstrations focus predominantly on 2D image recovery. For individual patches the proposed method requires over 100 seconds, and also the authors indicate that full image recovery could take an impractical amount of time (hundreds of hours). While DeepRD may offer a solution, the real-world applicability of RDM, especially for video

and 3D data, appears limited. The effectiveness of RDM on data with non-symmetric aberrations along the z-axis in 3D data is also questionable.

RDM is primarily focused on 2D image recovery but can be easily extended to higher dimensions, namely video data as shown in the manuscript for the miniscope system and for 3D deconvolution, as shown in our new light-sheet fluorescence microscopy example. There has been a misunderstanding here, **ring deconvolution runs on full images (512x512) in about 60 seconds** and DeepRD does so in about a tenth of a second. The statement about 'impractical amount of time (hundreds of hours)' is if one attempts to solve the rigorous superposition integral without using the LRI assumption, **not for RDM**. We realize that this line may be confusing, and so we remove it from the paper. Instead, we clarify with following language:

“Ring deconvolution provides the best reconstruction, but it is also the largest and slowest of the methods tested, taking about 60 seconds for a full image. However, if we were to try to deblur the image rigorously without using RDM, it would take hundreds of hours, despite being theoretically equivalent to ring deconvolution. Thus ring deconvolution allows for relatively fast, accurate deblurring where it was once infeasible.”

Therefore, we feel that RDM has strong real-world applicability to 3D and video data as evidenced by our waterbear videos (about 1 hour reconstruction time for a full video) and LSFM reconstructions (about 7 minutes reconstruction time per volume). However, we expect these speeds to get dramatically faster over time since the RDM algorithms are extremely parallelizable. In fact, early experiments using multiple GPUs with the Jax framework has shown a 20-fold decrease in runtime for ring deconvolution. This means that ring deconvolution on a 1024x1024 image would take about a second.

Runtime of ring deconvolution in iterations per second as a function of the number of GPUs used. Experiments were run using Chromatix <https://github.com/chromatix-team/chromatix>.

For the general case of aberrations along z , RDM can simply be applied slice-by-slice, using new PSFs for each new z position. The exact calibration mechanism is dependent on the type of z -dependent aberrations, but if it were just defocus, then only a single Seidel coefficient would need to be updated per z -slice. However, if the 3D PSF is shift-invariant along the z -dimension, then it is possible to incorporate RDM for 3D deconvolution just like we did for sheet deconvolution in the new LSFM experiment. This would involve inverting a 3D version of ring convolution, which would amount to convolving cylindrical regions of the sample volume with cylindrical regions of the sample PSF. We leave this as a future direction.

- The experimental results provided do not sufficiently highlight RDM's advantages over existing methods. A more thorough comparison with various PSF approximations (e.g., simple gaussian kernel), deconvolution algorithms (especially spatially-varying ones), and under different noise conditions (to see how robust the proposed method is) is advisable. The selection of microscopy modalities for demonstration seems arbitrary, and their relevance to spatially varying aberration should be clarified.
- In addition, a comparison with conventional deconvolution algorithms using a standard Gaussian PSF kernel would be useful.

We agree that a more thorough comparison with existing methods would benefit the paper. Consequently, we include a new experiment which compares RDM both quantitatively and qualitatively against Gaussian kernel deconvolution, spatially-varying modal deconvolution, and spatially-varying Richardson Lucy (SVRL) from Turcotte et al. We run these comparisons on an experimental grid of beads from the micro-endoscopy multimode fiber system (see above, and Appendix A). This experiment allows us to use these other spatially-varying methods since a rigorous grid calibration was provided for this system (i.e., a PSF at every point in the FoV). It also serves as a quantitative test bed since we are able to measure the full width half maxima of the beads after reconstruction with each method. RDM outperforms all the other methods we test in this regard.

In addition to the deblurring performance, these other spatially-varying methods all require extensive and specific calibration (e.g., an equally-spaced grid of PSFs). In contrast, RDM can be run with a single, randomly generated calibration image.

The selection of microscope modalities was intended to cover a wide range of systems with different underlying mechanisms, but that all suffered spatially-varying aberrations. In the beginning of each subsection in Results we discuss why each of these systems is subject to spatial-variance. The Miniscope allowed us to show how RDM can improve systems that are constrained to simple, uncorrected optics by their application (i.e., miniature *in vivo* imaging). The high NA multicolor microscopy shows how RDM can push the FoV of more commonly-used, commercial systems and applies to multicolor

imaging. The micro-endoscope system shows how RDM can significantly improve even complex, non-traditional imaging as long as it preserves symmetry and demonstrates that RDM works equally well on point-scanning systems. Finally, the LSFM shows how RDM can extend to symmetries beyond rotational symmetry and can improve aberrations induced by illumination, not just by optics. We add the following statement to the paper:

“We demonstrate this on four diverse microscope modalities: miniature microscopy, multicolor fluorescence microscopy, multimode fiber micro-endoscopy, and light-sheet fluorescence microscopy. These modalities each contain different characteristics and imaging mechanisms that are representative of a large range of systems, thereby forming a comprehensive basis to demonstrate the wide applicability and practical relevance of RDM.”

- The manuscript is well-written, but its structure could be improved for better clarity, particularly in the sections detailing reconstruction algorithms and microscopy modalities. A more seamless transition between sections is suggested.

We appreciate the feedback and have introduced a few minor improvements throughout the manuscript for added clarity including the statement from the previous bullet point.

- The claim in Figure 6 that elements 5 and 6 are not resolvable by other methods needs verification, which I think not true.

We agree that the statement is too vague as written and have corrected it to:

“In the first row, ring deconvolution and DeepRD clearly resolve resolution target elements near the edges of the FoV, which are not as well resolved by the two other methods.”

- Consider revising the title better to reflect the focus on rotationally symmetric imaging systems.

We appreciate this suggestion and have changed the title of the paper to:

“Ring Deconvolution Microscopy: exploiting symmetry for efficient spatially-varying aberration correction”

- The results from real data are less convincing compared to those from simulated scenarios.

This is an expected trend for all computational methods and is largely due to model mismatch. In simulation, we assume systems are perfectly rotationally symmetric, but in

reality, misalignment, manufacturing errors, noise correlations, and other experimental sources of error lead to only approximate rotational symmetry. Consequently, we expect some degree of degradation from simulation to experiment, but still find that RDM produces useful experimental results that match the trends seen in simulation.

- An explanation of why the chosen five Seidel coefficients are adequate is necessary, especially in light of Figure 6, which shows persistent artifacts from Seidel deconvolution. Can additional coefficients offer improvements for these cases?

The artifacts in Figure 6 are due to noise, not Seidel deconvolution. The artifacts are even worse in the baseline deconvolution, which uses an experimental PSF. Having said that, it is true that only using five Seidel coefficients induces some approximation error.

We add a new experiment and section to the paper regarding higher-order coefficients (see above and Appendix C.1). In summary, we find that the primary coefficients perform well empirically in approximating higher-order aberrations. Our codebase allows users to easily change the number of Seidel coefficients used in the approximation for their specific use-case (we have found this does not significantly improve RDM's performance).

- Detailed information on the running time of the proposed algorithm under various conditions should be provided to assess its practicality.

All runtimes are subject to the computational resources at hand. We detail ours in the Computation subsection (4.6): computation was done using Python on a single GPU, either a NVIDIA GeForce RTX 3090 or NVIDIA RTX A6000. 512 x 512 images with ring deconvolution take about 60 seconds to deblur a full image. In the Miniscope and high NA microscopy experiments, ring deconvolution took ~115 seconds for a 1024x1024 image. In the multimode fiber experiment, ring deconvolution takes about ~3 seconds for a 120x120 image and ~20 seconds for a 360x360 image. Finally in the LSFM experiment, sheet deconvolution took about 7 minutes for a 512x512x160 volume. We add these details to Sec. 4.6.

OVERVIEW

We thank the editor and reviewers for their support of our manuscript and are grateful that the improvements made in the last review cycle have been recognized.

To address the remaining technical feedback, we have made the following changes to the manuscript:

- Added a new appendix, **Appendix B**, which quantifies the performance of our light-sheet calibration procedure and sheet deconvolution. This includes a new figure copied below, **Figure 9**

Figure 9: Quantifying Sheet deconvolution microscopy. a) Error evaluation of PSF fitting models. N=6 image stacks of randomly-scattered point sources are acquired with a light-sheet microscope and fit using either with the modified Gibson-Lanni model, by interpolating the measured PSFs, or by using the (denoised) center PSF and assuming shift-invariance. The PSNRs between the generated PSFs and the measured PSFs from the remaining calibration image stacks is computed for all three PSF models. These PSNRs are stratified by regions split along the light-sheet spread direction; center is where the light-sheet is the thinnest, while edge is where the light-sheet is the most spread. The resulting average PSNRs and their standard deviations in each region are plotted as a function of region. b) Forward model computation time for different data sizes. The runtimes of three different light-sheet forward models are plotted as a function of data size; true blur (manually superimposing the shift-varying PSFs at every point in the 3D volumes), sheet convolution, and standard 3D convolution. Times are acquired by processing randomly-generated 3D volumes and are averaged over 10 trials. c) Quantitative evaluation of sheet deconvolution. N=20 3D volumes of randomly sized and oriented ellipsoids are generated. These objects are blurred via true blur using PSFs generated from our modified Gibson-Lanni model, and are corrupted with Poisson noise (SNR 15) to simulate light-sheet measurements. The simulated measurements are deblurred with sheet deconvolution, iterative gradient-based deconvolution with total variation regularization, Richardson-Lucy deconvolution, and Wiener deconvolution. The process is repeated with increasing values of the β spread parameter (see 4.4.2), beginning with a system that is shift-invariant ranging to a system which is highly shift-variant due to rapid light-sheet spread. The average PSNR over all 20 objects for each method is plotted as a function of the spread parameter.

- Expanded the discussion to point towards future work in dynamic imaging.
- Made several minor organizational revisions and clarifications to language.

Below, we respond to the comments from each reviewer in turn.

Reviewer #1

R1: The proposed method is well-motivated, theoretically justified, and broadly applicable. I expect it will be widely used in microscopy and other imaging applications.

The revision is a significant improvement over the original submission. I especially appreciate the new light sheet results. All my concerns have been addressed and I support publication.

We appreciate the reviewer's support.

Reviewer #2

R2: The authors have addressed most of my concerns... I suggest accept the manuscript.

We appreciate the reviewer's support and respond to the remaining suggestions below.

R2: In this light-sheet fluorescence microscopy (LSFM) experiment, the authors applied Ring Deconvolution to address the issue of spatially varying aberrations in light-sheet microscopy. Since the thickness of the light sheet varies along the propagation axis (typically the z-axis within the field of view), causing resolution degradation at the edges, the authors effectively restored the resolution across the entire field of view, particularly in the edge regions, using ring deconvolution. The experimental details of the PSF calibration process could be appropriately supplemented, such as how to select point source locations, handle noise interference, and perform interpolation or fitting to generate the 3D PSF. Additionally, error evaluation should be included to verify the accuracy of the generated PSF.

Thanks for the suggestions, we address them as follows:

We have supplemented Section 4.4.2 to discuss additional information about the PSF calibration process. The new text is below:

In order to deploy the above models experimentally, one must ensure that the point sources are sufficiently sparse such that the PSFs are mostly non-overlapping. The exact point source locations are not important and can be estimated. Noisy PSFs are best handled with the Gibson-Lanni fitting method, which uses synthetic PSFs but still fits the measured PSFs well (see Appendix B). The interpolation method is more sensitive to noise but can be denoised effectively with thresholding and median filtering. The fitting procedures are detailed in our codebase, and follow the same general structure: first take a calibration stack of randomly-scattered point sources, estimate the locations of the PSFs using local peak finding algorithms, produce generated PSFs at those locations, and use the error between the generated and measured PSF to update the PSF model. In the case of interpolation, the last

step is replaced by linearly interpolating the two closest measured PSFs to the desired location to create the generated PSF there.

We verified the accuracy of the generated LFSM PSFs across the field-of-view. This is shown in the newly added Appendix B and Figure 9 (see main response).

R2: The analysis of Sheet Deconvolution's performance in light-sheet fluorescence microscopy is insufficiently detailed. While the experimental results highlight its resolution improvement and computational efficiency advantages, there is a lack of quantitative comparisons with standard 3D deconvolution methods. Metrics such as signal-to-noise ratio (SNR) and specific computation time or complexity analyses for different data sizes should be included to provide a more comprehensive evaluation.

We added a report on runtime over different data sizes and compared sheet deconvolution to standard 3D deconvolution techniques including iterative total variation deconvolution, Richardson-Lucy, and Wiener filtering. This is shown in the newly added Appendix B and Figure 9 (see above).

R2: I still suggest the authors discuss the validity of their methods under dynamic conditions (e.g., the two papers published on Nature Photonics 2023, 17, 679-687 and Nature Photonics 2015, 9, 529-535), and provide valuable insights into the versatility of these methods. They don't need to do experiments, but give some discussion.

We have added a discussion of the applicability of our methods in dynamic settings to the Discussion section of our manuscript. See the new text below:

RDM is well-suited to dynamic conditions in which the system or sample is changing, as long as the system calibration can be updated accordingly. For simpler cases that can be directly modeled, such as the time-varying deformation of an imaging fiber, it should be possible to update the initial calibrated PSFs with a theoretical model ([55,56]).

Reviewer #3

R3: The authors have made significant improvements compared to the previous version. The discussion on how RDM differs from related work and the additional results on various microscopy modalities are particularly helpful.

We thank the reviewer for their positive assessment. Our responses to the additional observations and comments are below.

R3: The advantages of the new concept of ring convolution are still not clearly presented. The visual differences between deconvolved images from RDM/Seidel/Sheet deconvolution and conventional methods seem minor.

The main benefit of RDM is improved image recovery in the presence of radially varying aberrations. Figure 5 gives a quantitative explanation of this improvement; see Figure 5 (e.g. deconvolution has a PSNR of 19, while RDM has a PSNR of 24).

On real data, although the PSNR/MSE cannot be calculated, our reconstructions do appear to be consistently better than deconvolution; the insets we have placed on Figures 1-7 make these improvements visible to the naked eye. The degree of improvement is up to the reader, but, we believe Figure 2 clearly shows a noticeably better reconstruction of an airforce target, rabbit liver, and live tardigrade.

R3: The necessity of DeepRD needs further justification because:

1. its performance appears to be significantly worse than that of ring deconvolution; and
2. the claimed faster processing time, which is a major motivation, is not demonstrated. It would be more convincing if this more advanced version offered additional advantages (I understand the interpretability of DeepRD is briefly discussed).

The faster processing time of DeepRD is demonstrated in Figure 5(i). (It is possible this was missed due to a typo on our end.) We have fixed the typo and included the adjusted figure below.

Here, it is shown that DeepRD is almost three orders of magnitude faster than RDM.

We have also added the following sentence in our discussion of speed:

DeepRD is almost three orders of magnitude faster than RDM.

R3: As an image restoration method for deconvolution microscopy, I still believe it is essential to comprehensively compare this approach with other state-of-the-art methods, such as CARE (15, 1090–1097 (2018)), even if the forward model might be slightly different. This will also assist in justifying the new concept of RDM.

We agree. The U-Net baseline in Figure 2 is the CARE baseline suggested by the reviewer. It is the CARE model trained on our data. The performance is worse than RDM.

R3: I cannot agree with the assertion that we should accept that the method generalizes “surprisingly” well for real data when trained on simulated data, as stated on page 6.

We removed this claim.

R3: The results obtained from the demo code on real data (<https://github.com/apsk14/rdmpy/blob/main/demo.ipynb>), last example, live tardigrade?) appear to exhibit serious artifacts that do not represent true tissues after ring deconvolution. These results also differ from those presented in the paper. Maybe there are some parameters to be tuned?

The code is very well-structured, making it easy to follow and understand. However, as mentioned, the results obtained from the demo code on real data (demo.ipynb) appear to exhibit serious artifacts that do not represent true tissues after ring deconvolution and DeepRD. These results also differ from those presented in the paper.

Thank you for going through the code in detail! The section you reference in the demo notebook is actually using a different sample from the one shown in the paper. It is worth noting also that the sample shown in the demo notebook has significant amounts of algae in it (in addition to the tardigrade), which may look like artifacts but is in fact part of the sample. For additional clarity, we added new cells to the notebook which run ring deconvolution and DeepRD on the same sample that we use in the paper; as expected, the output matches exactly the images presented in the paper.

R3: On page 18, the phrase “with a final density 5×10^{-4} of the stock” should be corrected to “ 5×10^{-4} ”.

Thank you; we fixed the typo.

R3: The distinction between sheet deconvolution and ring deconvolution is noted, but it is unclear why this should be included in a paper focused on ring deconvolution microscopy, even though the theory of rotational symmetry may be relevant.

Thank you for the thoughtful point. We view sheet deconvolution as a natural and practical extension of ring deconvolution.